# LINEAR SEPARABILITY IN CONTRASTIVE LEARNING VIA NEURAL TRAINING DYNAMICS

## ABSTRACT

The SimCLR method for contrastive learning of invariant visual representations has become extensively used in supervised, semi-supervised, and unsupervised settings, due to its ability to uncover patterns and structures in image data that are not directly present in the pixel representations. However, this success is still not well understood; neither the loss function nor invariance alone explains it. In this paper, we present a mathematical analysis that clarifies how the geometry of the learned latent distribution arises from SimCLR. Despite the nonconvex SimCLR loss and the presence of many undesirable local minimizers, we show that the training dynamics driven by gradient flow tend toward favorable representations. In particular, early training induces clustering in feature space. Under a structural assumption on the neural network, our main theorem proves that the learned features become linearly separable with respect to the ground-truth labels. To support our theoretical insights, we present numerical results that align with our theoretical predictions.

## 1 INTRODUCTION

Unsupervised learning of effective representations for data is one of the most fundamental problems in machine learning, especially in the context of image data. The widely successful *discriminative approach* to learning representations of data is most similar to fully supervised learning, where features are extracted by a backbone convolutional neural network, except that the fully supervised task is replaced by an unsupervised or *self-supervised* task that can be completed without labeled training data.

Many successful discriminative representation learning methods are based around the idea of finding a feature map that is *invariant* to a set of transformations (i.e., data augmentations) that are expected to be present in the data. For image data, the transformations may include image scaling, rotation, cropping, color jitter, Gaussian blurring, and adding noise, though the question of which augmentations give the best features is not trivial (Tian et al., 2020). Invariant feature learning methods include VICReg Bardes et al. (2021), Bootstrap Your Own Latent (BYOL) (Grill et al., 2020), Siamese neural networks Chicco (2021), and contrastive learning techniques such as SimCLR Chen et al. (2020) (see also (Hadsell et al., 2006; Dosovitskiy et al., 2014; Oord et al., 2018; Bachman et al., 2019)).

In contrastive learning, the primary self-supervised task is to differentiate between positive and negative pairs of data instances. The goal is to find a feature map for which positive pairs have maximally similar features, while negative pairs have maximal different features. The positive and negative examples do not necessarily correspond to classes. In SimCLR, positive pairs are images that are the same up to a transformation, while all other pairs are negative pairs. Contrastive learning has also been successfully applied in supervised (Khosla et al., 2020) and semi-supervised contexts (Li et al., 2021; Yang et al., 2022; Singh, 2021; Zhang et al., 2022b; Lee et al., 2022; Kim et al., 2021; Ji et al., 2023), and has been used for learning Lie Symmetries of partial differential equations Mialon et al. (2023) (for a survey see Le-Khac et al. (2020)).

All invariance based feature extraction techniques must address the fundamental problem of dimension collapse, whereby a method learns the trivial constant map $f(x) = c$ (or a very low rank map), which is invariant to *all* transformations, but not informative or descriptive. There are various ways to prevent dimension collapse. In contrastive learning the role of the negative pairs is to prevent collapse by creating repulsion terms in the latent space, however, full or partial collapse can still occur (Jing

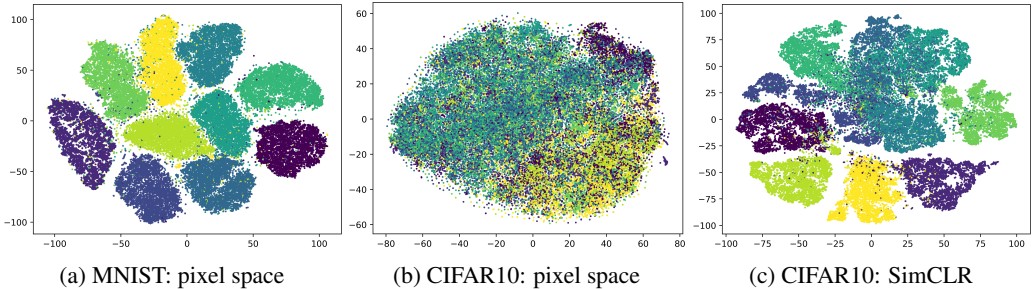

| (a) MNIST: pixel space | (b) CIFAR10: pixel space | (c) CIFAR10: SimCLR |

Figure 1: t-SNE visualizations of the MNIST and CIFAR10 data sets. In (a) and (b) the images are represented by the raw pixels, while (c) gives a visualization of the SimCLR embedding. This illustrates how SimCLR is able to uncover clustering structure in data sets.

et al., 2021; Zhang et al., 2022a; Shen et al., 2022; Li et al., 2022). In BYOL collapse is prevented by halting backpropagation in certain parts of the loss, and incorporating temporal averaging. In VICReg, additional terms are added to the loss function to maintain variance in each latent dimension, as well as to decorrelate variables.

Provided dimensional collapse does not occur, a fundamental unresolved question surrounding many feature learning methods is: why do they work so well at producing embeddings that uncover key features and patterns in data sets? As a simple example, consider fig. 1. In fig. 1a and fig. 1b we show t-SNE (Van der Maaten & Hinton, 2008) visualizations of the MNIST (Deng, 2012) and CIFAR10 (Krizhevsky et al., 2009) data sets, respectively, using their pixel representations. We can see that visual features are not required on MNIST, which is highly preprocessed, while for CIFAR10 the pixel representations are largely uninformative, and feature representations are essential. In fig. 1c we show a t-SNE visualization of the latent embedding of the SimCLR method applied to CIFAR10, which indicates that SimCLR has uncovered a strong clustering structure in CIFAR10 that was not present in the pixel representation.

The goal of this paper is to provide a framework that can begin to address this question, and in particular, to explain fig. 1. To do this, we assume the data follows a *corruption* model, where the observed data is derived from some clean data with distribution $\mu$ that is highly structured or clustered in some way (e.g., follows the manifold assumption with a clustered density). The observed data is then obtained by applying transformations at random from a set of augmentations $\mathcal{T}$ to the clean data points (i.e., taking different views of the data), producing a corrupted distribution $\tilde{\mu}$. The main question that motivated our work is that of understanding what properties of the original clean data distribution $\mu$ can be uncovered by unsupervised contrastive feature learning techniques? That is, once an invariant feature map $f : \mathbb{R}^D \to \mathbb{R}^d$ is learned, is the latent distribution $f_{\#}\tilde{\mu}$ similar in any to the clean distribution $\mu$, or can it be used to deduce any geometric or topological properties of $\mu$?

The main contribution of our paper can be summarized as follows.

> **Contribution:** Despite the presence of many undesirable local minimizers of the SimCLR loss, we show, using a neural-kernel analysis, that the interaction between the data distribution and the augmentation shapes the gradient dynamics and drives the learned representation. In particular, cluster structure in $\mu$ can persist and sharpen in the learned features.

Our work provides a framework for explaining the success of SimCLR and other invariance-based feature learning technique (for simplicity, we focus on SimCLR, and indicate in the appendix how our results extend to other methods). Our work complements research on dimension collapse in contrastive learning (Jing et al., 2021; Zhang et al., 2022a; Shen et al., 2022; Li et al., 2022), as our findings hold even without collapse. We also highlight recent work (Meng & Wang, 2024) on the training dynamics of contrastive learning through a continuum limit PDE. Other related works, such as (HaoChen et al., 2021; Balestriero & LeCun, 2022), provide guarantees for downstream tasks like semi-supervised learning by studying the alignment between class-membership clusters and an "augmentation graph." Our paper complements these by examining when this alignment holds in contrastive learning.

**Outline:** In section 2 we provide an overview of contrastive learning and introduce our data corruption model. In section 3 we derive and study the optimality conditions for the SimCLR loss and characterize its stationary points. In section 4 we analyze the neural training dynamics of SimCLR and present the main result: under mild conditions on the parameters, the learned representations become linearly separable.

## 2 CONTRASTIVE LEARNING

We describe here our model for corrupted data in the setting of contrastive learning, and a reformulation of the SimCLR loss that is useful or our analysis. Let $\mu \in \mathbb{P}(\mathbb{R}^D)$ be a data distribution in $\mathbb{R}^D$. Let $\mathcal{T}$ be a set of transformation functions $T : \mathbb{R}^D \to \mathbb{R}^D$ that is measurable such that, for a given $x \in \mathbb{R}^D$, $T(x) \in \mathbb{R}^D$ represents a perturbation of $x$, such as a data augmentation (e.g., cropping and image, etc.). Let $\tilde{\mu} \in \mathbb{P}(\mathbb{R}^D)$ denote the distribution obtained by perturbing $\mu$ with the perturbations defined in $\mathcal{T}$. That is, we choose a probability distribution $\nu \in \mathbb{P}(\mathcal{T})$ over the perturbations, and samples from $\tilde{\mu}$ are generated by sampling $x \sim \mu$ and $f \sim \nu$, and taking the composition $f(x)$.

We treat $\mu$ as the original clean data, which is not observable, while the perturbed distribution $\tilde{\mu}$ is how the data is presented. Our goal is to understand whether contrastive learning can recover information about the original data $\mu$, provided the distribution of augmentations $\nu$ is known.

Ostensibly, the objective of contrastive learning is to identify an embedding function $f : \mathbb{R}^D \to \mathbb{R}^d$ that is invariant to the set of transformations $\mathcal{T}$. Provided such an invariant map is identified, $f$ pushes forwards both $\mu$ and $\tilde{\mu}$ to the same latent distributions, that is

$$f_{\#}\tilde{\mu} = f_{\#}\mu.$$

As a result, the desirable map $f$ is not only invariant to perturbations from $\mathcal{T}$ but also successfully retrieves the unperturbed data $\mu$, ensuring that the embedded distribution $f_{\#}\mu$ serves as a pure feature representation of the given data. However, it is far from clear how $\mu$ and $f_{\#}\mu$ are related, and whether any interesting structures in $\mu$ (such as clusterability) are also present in $f_{\#}\mu$.

For instance, if $\tilde{\mu}$ represents image data, contrastive learning aims to discover a feature distribution $f_{\#}\tilde{\mu}$ that remains invariant to transformations such as random translation, rotation, cropping, Gaussian blurring, and others. As a result, this feature distribution effectively captures the essential characteristics of the data without being influenced by these perturbations. These feature distributions are often leveraged in downstream tasks such as classification, clustering, object detection, and retrieval, where they achieve state-of-the-art performance (Le-Khac et al., 2020).

To achieve this, a cost function is designed to bring similar points closer and push dissimilar points apart through the embedding map, using attraction and repulsion forces. A popular example is the Normalized Temperature-Scaled Cross-Entropy Loss (NT-Xent loss) introduced by Chen et al. (2020), which leads to the optimization problem

$$\min_{f:\mathbb{R}^D \to \mathbb{R}^d} \mathbb{E}_{x \sim \mu, T, T' \sim \nu} \log \left( 1 + \frac{\sum_{h \in \{T, T'\}} \mathbb{E}_{y \sim \mu} \left[ \mathbb{1}_{x \neq y} \exp \left( \frac{\mathrm{sim}_f(T(x), h(y))}{\tau} \right) \right]}{\exp \left( \frac{\mathrm{sim}_f(T(x), T'(x))}{\tau} \right)} \right), \qquad (1)$$

where $\nu \in \mathbb{P}(\mathcal{T})$ is a probability distribution on $\mathcal{T}$, which is assumed to be a measurable space, $\tau$ is a given parameter, and $\mathrm{sim}_f : \mathbb{R}^D \times \mathbb{R}^D \to \mathbb{R}$ is a function measuring the similarity between two embedded points with $f$ in $\mathbb{R}^d$ defined as $\mathrm{sim}_f(x, y) = \frac{f(x) \cdot f(y)}{\|f(x)\| \|f(y)\|}$. The denominator inside the log function acts as an attraction force between perturbed points from the same sample $x$, while minimizing the numerator acts as a repulsion force between points from different samples $x$ and $y$. Thus, the minimizer $f$ of the cost is expected to exhibit invariance under the group of perturbation functions from $\mathcal{T}$.

$$f(T(x)) = f(x), \qquad \forall x \in \mathbb{R}^D, \forall T \in \mathcal{T}. \qquad (2)$$

The repulsion force prevents dimensional collapse, where the map sends every sample to a constant: $f(x) = c$ for all $x \in \mathbb{R}^D$.

An important observation is that the NT-Xent loss becomes independent of the data distribution once the feature map $f$ is invariant, meaning that the latent distribution corresponding to an invariant minimizer may bear little resemblance to the input data. Interestingly, a similar effect is observed in

other unsupervised learning models such as VICReg (Bardes et al., 2021) and BYOL (Grill et al., 2020) (see the appendix for further discussion). Despite the fact that the NT-Xent loss has minimizers completely independent of the original data distribution $\mu$, in practice the clustering structure of $\mu$ often emerges in the latent space which results in the state-of-the-art performance in machine learning tasks.

To better understand this behavior, we analyze the NT-Xent loss by examining its minimizer and the dynamics of gradient descent. In order to overcome challenges caused by the nondifferentiability of the angular similarity $\text{sim}_f$ and the nonuniqueness of solutions (e.g., any $kf$ is also a minimizer for $k > 0$), we reformulate the loss to simplify the analysis. This leads to the generalized formulation of the NT-Xent loss in eq. (1).

**Definition 2.1.** The cost function we consider for contrastive learning is

$$\inf_{f \in \mathcal{C}} \left\{ L(f) := \mathop{\mathbb{E}}_{x \sim \mu, T, T' \sim \nu} \Psi \left( \frac{\mathbb{E}_{y \sim \mu} \, \eta_f(T(x), T'(y))}{\eta_f(T(x), T'(x))} \right) \right\}, \tag{3}$$

where $\Psi : \mathbb{R} \to \mathbb{R}$ is a nondecreasing function, $\mathcal{C}$ is a constraint set, and $\eta_f$ is defined as

$$\eta_f(x, y) = \eta(\|f(x) - f(y)\|^2 / 2), \tag{4}$$

where $\eta : \mathbb{R}_{\geq 0} \to \mathbb{R}$ is a differentaible similarity function that is maximized at $0$.

The formulation in eq. (3) generalizes the original formulation in eq. (1) by removing the indicator function $\mathbb{1}_{x \neq y}$, as the effect of this function becomes negligible when a large $n$ is considered. Furthermore, the generalized formulation introduces a differentaible similarity function. This simplifies the analysis of the minimizer in the variational formulation. The generalized formulation can easily be related to the original cost function in eq. (1) by setting $\Psi(t) = \log(1 + t)$, $\eta(t) = e^{-t/\tau}$ and defining $\mathcal{C} = \{f : \mathbb{R}^D \to \mathbb{S}^{d-1}\}$. Then, the similarity function $\eta_f$ retains the same interpretation as angular similarity $\text{sim}_f$. This is because, if $f$ lies on the unit sphere in $\mathbb{R}^d$, and so

$$\exp\left(-\frac{1}{2\tau}\|f(x) - f(y)\|^2\right) = \exp\left(\frac{1}{\tau}(f(x) \cdot f(y) - 1)\right) = C \exp\left(\frac{1}{\tau} \frac{f(x) \cdot f(y)}{\|f(x)\|\|f(y)\|}\right),$$

where $C = \exp(-1/\tau)$. The consideration of the constraint also resolves the issue in eq. (1), where $kf$, for any $k \in \mathbb{R}$, could be a minimizer of eq. (1). Thus, in the end, the introduced formulation in eq. (3) remains fundamentally consistent with the original NT-Xent cost structure.

## 3 OPTIMALITY CONDITION

In this section, we aim to find the optimality condition for eq. (3) and analyze properties of the minimizers. Using the first optimality condition we can characterize the minimizer of the NT-Xent loss in eq. (3). The following theorem describes the possible local minimizers of eq. (3), considering the constraint set defined as $\mathcal{C} = \{f : \mathbb{R}^D \to \mathbb{S}^{d-1}\}$.

**Theorem 3.1.** *Given a data distribution $\mu \in \mathbb{P}(\mathbb{R}^D)$, let $f \in \mathcal{C} = \{f : \mathbb{R}^D \to \mathbb{S}^{d-1}\}$ be an invariant map such that the embedded distribution $f_\# \mu$ is a symmetric discrete measure satisfying*

$$\int_{\mathbb{S}^{d-1}} h(x_1, y) df_\# \mu(y) = \int_{\mathbb{S}^{d-1}} h(x_2, y) df_\# \mu(y), \tag{5}$$

*for all $x_1, x_2 \sim f_\# \mu$ and for all anti-symmetric functions $h : \mathbb{S}^{d-1} \times \mathbb{S}^{d-1} \to \mathbb{S}^{d-1}$ such that $h(x, y) = -h(y, x)$. Then, $f$ is a stationary point of eq. (3) in $\mathcal{C}$.*

*Remark* 3.1. Examples of the embedded distribution $f_\# \mu$ in Theorem 3.1 include a discrete measure, $f_\# \mu = \frac{1}{n} \sum_{i=1}^n \delta_{x_i}$, with points $x_i$ evenly distributed on $\mathbb{S}^{d-1}$, or all points mapped to a single point, $f_\# \mu = \{x\}$. Figure 2 shows loss plots for different embedded distributions, $f_\# \mu = \frac{1}{K} \sum_{i=1}^K \delta_{x_i}$, with points $x_i$ evenly distributed on $\mathbb{S}^1$, illustrating how each stationary point relates to the loss.

The first plot shows the loss decreasing with the number of clusters, leveling off after a certain point. The second plot shows the loss decreasing as the minimum squared distance between cluster points narrows, plateaus once a threshold is reached. Both suggest that increasing the number of clusters or using a uniform distribution on $\mathbb{S}^1$ minimizes the NT-Xent loss. Additionally, increasing the number of points and decreasing $\tau$ further reduces the loss. The third plot reveals a linear relationship between $\tau$ and the threshold for the minimum squared distance, offering insight into the optimal cluster structure for minimizing the loss at a given $\tau$.

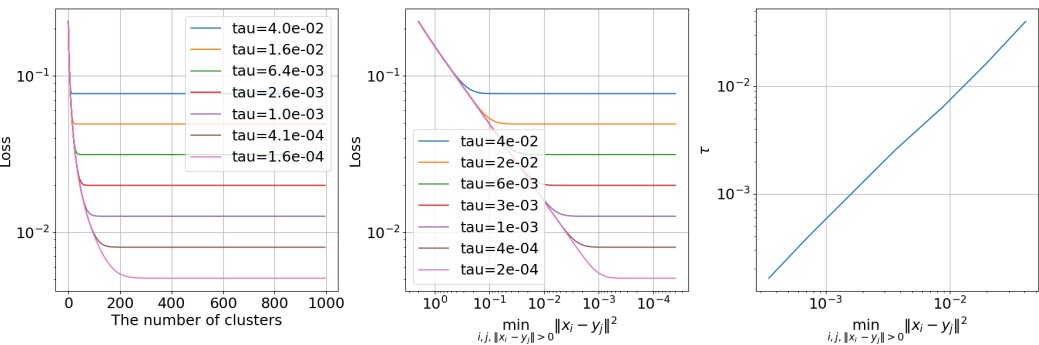

Figure 2: The figure shows the NT-Xent loss for different embedded distributions $f_{\#}\mu = \frac{1}{K}\sum_{i=1}^{K}\delta_{x_i}$ with $x_i$ on $\mathbb{S}^1$. The first plot shows the loss decreasing with the number of clusters, then plateauing. The second shows the loss decreasing with the minimum squared distance between cluster points, stopping at a threshold. Both suggest that increasing clusters and decreasing $\tau$ reduce the loss. The third plot shows a linear relationship between $\tau$ and the minimum distance.

*Remark* 3.2. Theorem 3.1 is related to the result from Wang & Isola (2020), where the authors studied local minimizers by minimizing the repulsive force under the assumption of an invariant feature map. They showed that, asymptotically, the uniform distribution on $\mathbb{S}^{d-1}$ becomes a local minimizer as the number of negative points increases. Our result extends this by offering a more general, both asymptotic and non-asymptotic characterization of local minimizers, broadening their findings.

It follows from Theorem 3.1 that gradient descent on the NT-Xent loss can lead to solutions that are *completely independent* of the original data distribution $\mu$. For instance, if $\mu$ has some underlying cluster structure, with multiple clusters, there are minimizers of the NT-Xent loss, i.e., an invariant map $f$, that map onto an *arbitrary* distribution in the latent space, completely independent of the clustering structure of $\mu$. However, in practice, when the map is parameterized using neural networks, and trained with gradient descent on $L(f)$, it is very often observed that the clustering structure of the original data distribution $\mu$ emerges in the latent space (see fig. 1). In fact, our results in section 4 show that this is true even if we initialize gradient descent very poorly, starting with an invariant $f$ mapping to the uniform distribution $\mathcal{U}(\mathbb{S}^{d-1})$!

Although the contrastive loss $L(f)$ has minimizers that ignore the data distribution $\mu$, leading to poor results, contrastive learning often achieves excellent performance in practice. This suggests that the neural network's parameterization and gradient descent optimization are *selecting* a good minimizer for $L(f)$, producing well-clustered distributions in the latent space. To understand this, we will analyze the dynamics of neural network optimization during training in the following sections.

## 4 OPTIMIZATION OF NEURAL NETWORKS

Here, we study contrastive learning through the lens of the associated neural network training dynamics, which illustrates how the data distribution enters the latent space through the neural kernel. In this section, we use the notation $[\![n]\!] = \{1, \ldots, n\}$.

### 4.1 GRADIENT FLOW FROM NEURAL NETWORK PARAMETERS

Let $w \in \mathbb{R}^m$ be a vector of neural network parameters, $\{x_1, \ldots, x_n\} \subset \mathbb{R}^D$ be data samples, and $f(w, x_i) = (f^1(w, x_i), \ldots, f^d(w, x_i))^\top \in \mathbb{R}^d$ be an embedding function where each function $f^k : \mathbb{R}^{n+D} \to \mathbb{R}$ is a scalar function for $k = 1, \ldots, d$. Consider a loss function $L = L(y, x) : \mathbb{R}^{d+D} \to \mathbb{R}$ with respect to $w$:

$$\mathcal{L}(w) = \frac{1}{n}\sum_{i=1}^{n} L(f(w, x_i), x_i). \tag{6}$$

Let $w(t)$ be a vector of neural network parameters as a function of time $t$. The gradient descent flow can be expressed as

$$\dot{w}(t) = -\nabla\mathcal{L}(w).$$

Due to the highly non-convex nature of $\mathcal{L}$, this gradient flow is difficult to analyze. By shifting the focus to the evolution of the neural network's output on the training data over time, rather than the weights, we can derive an alternative gradient flow with better properties for easier analysis. The following proposition outlines this gradient flow derived from the loss function $\mathcal{L}$. The proof of the proposition is provided in the appendix.

**Proposition 4.1.** *Let $w(t)$ be a vector of neural network parameters as a function of time $t$. Consider a set of data samples $\{x_1, \ldots, x_n\}$. Define $z_i(t) = f(w(t), x_i)$ for each $i \in [\![n]\!]$. Then, $z_i(t)$ satisfies the following ordinary differential equation (ODE):*

$$\dot{z}_i(t) = -\frac{1}{n}\sum_{j=1}^{n} K_{ij}(t)\nabla_z L(z_j(t), x_j), \tag{7}$$

*where the kernel matrix $K_{ij} \in \mathbb{R}^{d \times d}$ is given by*

$$(K_{ij}(t))^{kl} = K_{ij}^{kl}(t) = (\nabla_w f^k(w(t), x_i))^{\top}(\nabla_w f^l(w(t), x_j)). \tag{8}$$

*Remark* 4.1. We remark that the viewpoint in proposition 4.1, of lifting the training dynamics from the neural network weights to the function space setting, is the same that is taken by the Neural Tangent Kernel (NTK) Jacot et al. (2018). The difference here is that we do not consider an *infinite width* neural network, and we evaluate the kernel function on the training data, so the results are stated with kernel matrices that are data dependent (which is important in what follows). In fact, it is important to note that proposition 4.1 is very general and holds for any parameterization of $f$, e.g., we have so far not used that $f$ is a neural network.

*Remark* 4.2. The training dynamics in the absence of a neural network can be expressed as

$$\dot{z}_i(t) = -\nabla_z L(z_i(t), x_i) \tag{9}$$

where $K_{ij}$ is set to be identity matrices. In contrast to eq. (7), the above expression shows that the training dynamics on the $i$-th point $z_i$ are influenced solely by the gradient of the loss function at $x_i$, and there is no mixing of the data via the neural kernel $K$ (since here it is the identity matrix).

Using Proposition 4.1, we can analyze the invariance-preserving properties (and possible failures) of gradient descent, both without and with the neural network kernel. The following theorem compares the vanilla gradient flow with the gradient flow induced by the neural network kernel matrix.

**Theorem 4.2.** *Consider the gradient descent iteration from a gradient flow without a neural network in eq. (9), where $z_i^{(b)} = f(w^{(b)}, x_i)$ for all $i \in [\![n]\!]$, and*

$$z_i^{(b+1)} = z_i^{(b)} - \sigma\nabla_z L(z_i^{(b)}, x_i), \tag{10}$$

*with $\sigma$ as the step size. If $f(w^{(0)}, \cdot)$ is invariant to perturbations from $\nu$, as defined in eq. (2), then $f(w^{(b)}, \cdot)$ remains invariant for all gradient descent iterations.*

*On the other hand, in the case of a gradient descent iteration from eq. (7),*

$$z_i^{(b+1)} = z_i^{(b)} - \frac{\sigma}{n}\sum_{j=1}^{n} K_{ij}^{(b)}\nabla_z L(z_j^{(b)}, x_j), \tag{11}$$

*the invariance of $f$ at the $(b+1)$-th iteration holds only if $f$ is invariant at the $b$-th iteration and additionally satisfies the condition $\nabla_w f(w^{(b)}, T(x)) = \nabla_w f(w^{(b)}, x)$ for all $x \in \mathbb{R}^D$ and $T \in \mathcal{T}$.*

Theorem 4.2 contrasts optimization with and without neural networks. In standard gradient descent (*eq.* (10)), the map $f$ remains invariant if it is initially invariant. In contrast, with the neural kernel in *eq.* (11), even if $f$ starts invariant, an additional condition on $\nabla_w f$ is needed to maintain invariance. Since this condition is not guaranteed to be satisfied throughout the iterations, the optimization can cause $f$ to lose invariance, resulting in different dynamics compared to standard gradient descent.

Many other works have shown that the neural kernel imparts significant changes on the dynamics of gradient descent. For example, Xu et al. (2019a;b) established the frequency principle, showing that the training dynamics of neural networks are significantly biased towards low frequency information, compared to vanilla gradient descent.

## 4.2 Emergence of Linear Separability from Neural Network Gradient Flows

We explore how the neural network kernel $K_{ij}$ in eq. (8) influences the gradient flow induced by the contrastive learning loss. For simplicity, we assume that the embedding map $f$ takes values in $\mathbb{R}$, which allows us to express the gradient flow in eq. (7) as

$$\dot{z}(t) = -\frac{1}{n} K \nabla L,$$

where $z(t) = (z_1(t), \ldots, z_n(t))^\top \in \mathbb{R}^n$ denotes the feature representations at time $t$, $\nabla L = (\nabla_z L(z_1(t), x_1), \ldots, \nabla_z L(z_n(t), x_n))^\top$, and $K$ is the kernel matrix defined in eq. (8).

Our main theorem highlights how SimCLR can discover class structure from embeddings. With a mild, data-driven condition on the neural kernel matrix, namely a weak block structure in which, on average, intra-cluster interactions dominate inter-cluster ones along a contrast direction, and under explicit, verifiable bounds on the parameters and the initialization, gradient flow of the SimCLR loss yields linear separation in finite time. This identifies a concrete mechanism, quantifies the thresholds that trigger separation, and links training dynamics to the geometry of the learned representation.

For clarity, we state the result in a streamlined setting with two clusters and a one-dimensional embedding; the argument extends with minor modifications to multiple clusters and higher-dimensional embeddings. The full proof is given in the appendix.

**Theorem 4.3.** *Let $f : \mathbb{R}^p \times \mathbb{R}^D \to \mathbb{R}$ be such that, for each $x$, the map $w \mapsto f(w, x)$ is $C^2$. Let $X = \{x_i\}_{i=1}^n \subset \mathbb{R}^D$ be partitioned into clusters $X_1, X_2$. Assume:*

*(a) Initial closeness: $|f(w(0), x) - f(w(0), y)| \leq \varepsilon$ for all $x, y \in X$.*

*(b) Augmentation consistency: $\left| \mathbb{E}_{T \sim \nu}[f(w, T(x_i))] - \frac{1}{|X_q|} \sum_{j \in X_q} f(w, x_j) \right| < \gamma, \forall x_i \in X_q$.*

*(c) Gradient structure: with $g_i = \nabla_w f(w(0), x_i)$, define $\mu_q = \frac{2}{n} \sum_{i \in X_q} g_i$, $\xi_i = g_i - \mu_q$. Assume $\Theta = \|\mu_1 - \mu_2\|^2 > 0$, and $\frac{1}{n} \sum_i \|\xi_i\|^2 \leq \sigma^2$.*

*(d) Kernel stability: for $K_{ij}(t)$ defined in eq. (8), $\|K(t) - K(0)\|_{\mathrm{op}} \leq \delta$ for $t \in [0, T]$.*

*(e) Parameter regime: $\Theta \sqrt{n} \varepsilon \gg \frac{\varepsilon^3}{\tau} + \gamma + \sigma^2 + \delta$.*

*Then under the gradient flow of the simplified SimCLR loss*

$$L(z, x) = \mathbb{E}_{T \sim \nu} \left[ \log \frac{\frac{1}{n} \sum_{y \in X} \exp(-\|z - f(w, T(y))\|^2 / 2\tau)}{\exp(-\|z - f(w, T(x))\|^2 / 2\tau)} \right], \tag{12}$$

*there exists $t > 0$ such that $\{z_i(t)\}_{i=1}^n$ are linearly separable into clusters matching $X_1, X_2$.*

*Remark* 4.3. The loss formulation in eq. (12) is considered simplified because the data augmentation $T \sim \nu$ is applied only to one of the samples in each pairwise comparison, both in the numerator (repulsion) and the denominator (attraction). In contrast, the full SimCLR loss applies independent augmentations to both samples. This simplification preserves the essential structure of the contrastive objective while making the analysis more tractable.

*Remark* 4.4. The theorem is not restricted to neural networks. It applies to any family of parameterized functions for which the function is twice differentiable with respect to the parameters near the initialization. The result therefore covers generalized linear models, spline or radial basis parametrizations, and deep networks alike. The model enters only through its parameter gradients and the resulting kernel, which are the quantities constrained by the assumptions.

### 4.2.1 Verification of the parameter condition for linear separability.

In Theorem 4.3, we obtain a concrete parameter condition for the emergence of linear separation. The decisive quantity is the ratio between $\varepsilon$, the initial intra-cluster feature distance, and $\tau$, the temperature parameter. When $\varepsilon^3 / \tau$ is small, separation appears early in training. When this ratio is large, separation is delayed or may fail to occur.

To illustrate the role of the parameter condition in Theorem 4.3, Figure 3 compares training dynamics produced by the same neural network with one hidden layer of 100 neurons. The input dataset is

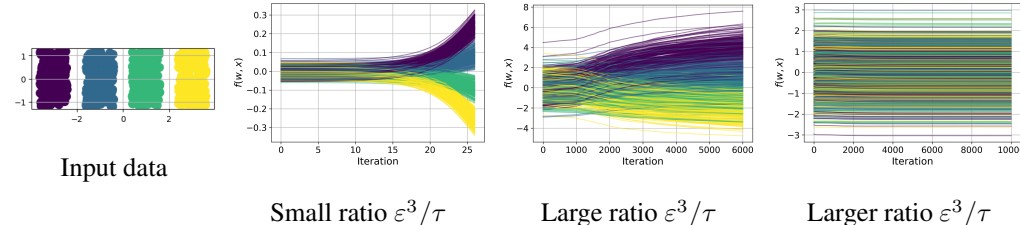

Input data

Small ratio $\varepsilon^3/\tau$        Large ratio $\varepsilon^3/\tau$        Larger ratio $\varepsilon^3/\tau$

Figure 3: This experiment verifies the parameter condition in Theorem 4.3. The training outcome is governed by the ratio between the initial intra-feature distance $\varepsilon$ and the temperature $\tau$ in the SimCLR loss. When $\varepsilon^3/\tau$ is small, the dynamics quickly produce linearly separable features after a short time. When $\varepsilon^3/\tau$ is large, separation is delayed and may not occur even after 10,000 iterations.

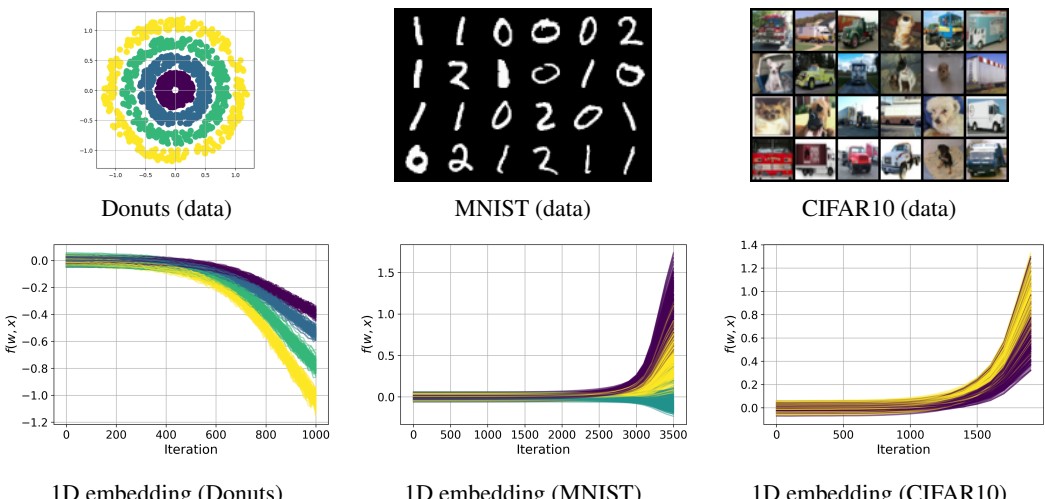

Donuts (data)        MNIST (data)        CIFAR10 (data)

1D embedding (Donuts)        1D embedding (MNIST)        1D embedding (CIFAR10)

Figure 4: Top row: visualizations of nonlinear input datasets that are not linearly separable. Bottom row: corresponding one-dimensional embedding trajectories over iterations under SimCLR gradient flow, showing increasing cluster structure and eventual linear separability.

clustered, as shown in the left panel. In this setting, condition (c) of Theorem 4.3 holds at initialization: if $f$ is $C^2$ in $w$, then $\nabla_w f(w(0), x)$ is locally Lipschitz in $x$. Thus, for points $x, x'$ within the same cluster of diameter at most $r$, $\|\nabla_w f(w(0), x) - \nabla_w f(w(0), x')\| \le Lr$, and the within-cluster deviations $\xi_i = g_i - \mu_{q(i)}$ satisfy $\frac{2}{n} \sum_{i \in X_q} \|\xi_i\|^2 \lesssim L^2 r^2$, while distinct clusters give $\mu_1 \ne \mu_2$. The second plot, corresponding to a small ratio $\varepsilon^3/\tau$ ($\varepsilon = 0.02$, $\tau = 0.1$), shows early linear separation. The third plot with ($\varepsilon = 1$, $\tau = 10^{-2}$) demonstrates delayed but eventual separation. The fourth plot, corresponding to a larger ratio ($\varepsilon = 1$, $\tau = 10^{-3}$), shows almost no change even after many iterations. These comparisons highlight the sensitivity of the training outcome to the interplay between the initial feature scale $\varepsilon$ and the temperature parameter $\tau$.

### 4.2.2 EMERGENCE OF LINEAR SEPARABILITY FROM COMPLEX DATASETS.

Figure 4 shows three experiments where features become linearly separable under gradient flow of the SimCLR loss, starting from datasets that are nonlinear and more complex than the well-clustered case in Figure 3. The networks are: a one-hidden-layer MLP with 100 neurons, a three-hidden-layer MLP with 100 neurons per layer, and a small CNN with two convolutional layers followed by one fully connected layer. The datasets are an artificial donut, MNIST (three classes), and CIFAR10 (two classes). Augmentations are as follows: for the donut, a random rotation in which $T(x)$ is sampled uniformly on the circle centered at the origin with radius $\|x\|$; for MNIST, Gaussian blur, random rotation, and resizing; for CIFAR10, Gaussian blur, random rotation, resizing, and color jittering. In all runs, features are initialized uniformly in an $\varepsilon$-ball and then evolved by gradient flow. Each curve traces one particle and is colored by its cluster label. Across all settings, the learned features become

increasingly linearly separable, indicating that the separation mechanism extends beyond the theory's simplifying assumptions. In particular, condition (c) in Theorem 4.3 may not hold at initialization for these complex datasets.

To connect these observations with the theorem, we examine the kernel $K$ in eq. (8). Condition (c) requires $K$ to be approximately block structured. Empirically, we observe that if this structure is absent at initialization, training first reshapes $K$ toward it, which typically adds iterations.

We visualize this transition in Figure 5 using two clusters so that the target pattern is a $2 \times 2$ block. Panels (a) and (b) show the linearly separable case from Figure 3 with two clusters. A clear block structure is already visible at iteration 0 because the data are well clustered (see Section 4.2.1), and it refines only modestly by iteration 5000. Panels (c) and (d) show the donut dataset, which is not well clustered: the initial kernel shows little separation, but by iteration 5000 a sharp block structure emerges. These visuals link the empirical separation in Figure 4 to the hypothesis in Theorem 4.3: once training sculpts $K$ into the required block form, linear separability follows.

A natural extension of Theorem 4.3 is a warm-start formulation. After some time $t_{\mathrm{b}} > 0$, once the kernel $K(t_{\mathrm{b}})$ exhibits the approximate block structure encoded in condition (c), replace $w(0)$ by $w(t_{\mathrm{b}})$ and redefine $g_i = \nabla_w f\big(w(t_{\mathrm{b}}), x_i\big)$, $\mu_q$, and $\xi_i$. Under the same parameter regime, the proof of Theorem 4.3 then applies verbatim and yields linear separability for $\{z_i(t)\}_{i=1}^n$ at some $t > t_{\mathrm{b}}$.

A complete dynamical theory that predicts when gradient flow sculpts $K(t)$ into the required block form is an independent and technically substantial problem. It complements the present contribution, which isolates a structural condition on $K$ and provides rigorous guarantees once that condition is met. A principled analysis of this kernel evolution, including how the choice of augmentation and neural network training influences it, is a promising direction for future work.

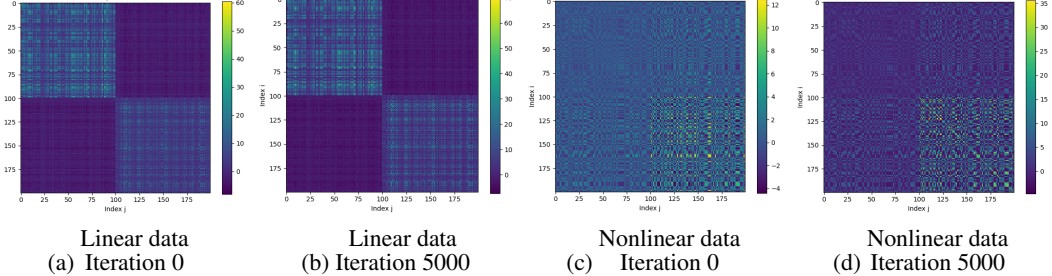

| Linear data | Linear data | Nonlinear data | Nonlinear data |
| (a) Iteration 0 | (b) Iteration 5000 | (c) Iteration 0 | (d) Iteration 5000 |

Figure 5: Evolution of the kernel matrix $K$ in eq. (8) for two-cluster datasets. Linear data: clear $2 \times 2$ block structure at $t = 0$ (a) and mild refinement by 5000 iterations (b). Nonlinear donut: no block structure at $t = 0$ (c) but pronounced blocks after 5000 iterations (d).

## 5 CONCLUSION AND FUTURE WORK

We study SimCLR through a variational analysis and through the gradient flow dynamics of a neural network that represents the embedding map. Our analysis shows that contrastive learning can induce linear separability under suitable conditions, as formalized in Theorem 4.3. Although we assume that the kernel is approximately block structured, the theorem provides explicit, verifiable conditions on the parameters and the initialization that ensure separation in finite time and offer practical guidance for their choice.

Future work includes developing a mean–field limit for the dynamics (Mei et al., 2018), analyzing the infinite–width regime in the NTK setting to obtain convergence guarantees and rates, and studying when training and data augmentations drive the kernel toward the required structure as observed in Figure 5, with the goal of relaxing the block assumption to weaker spectral conditions.

Additionally, in many contrastive learning studies, the neural network is trained end-to-end, but the last layer is discarded when extracting features. Prior works Bordes et al. (2023); Gui et al. (2023); Wen & Li (2022) have shown that this can improve feature quality. While we do not consider this effect in our current analysis, understanding its impact on training dynamics is another compelling direction for future investigation.

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

## A  APPENDIX

## B  APPENDIX

In the appendix, we present the proofs of that are missing in the main manuscript.

### B.1  INTERPRETATION OF VICREG AND BYOL

In this section, we further explore our observation that the NT-Xent loss becomes independent of the data distribution once the feature map $f$ is invariant. Consequently, the latent distribution corresponding to an invariant minimizer can be entirely unrelated to the input data. The following proposition demonstrates this.

**Proposition B.1.** *Suppose that $\mu \in \mathbb{P}(\mathbb{R}^d)$ is absolutely continuous and that the embedding map $f : \mathbb{R}^D \to \mathbb{R}^d$ is invariant under the distribution $\nu$ (satisfying eq. (2)). By applying a change of variables, we obtain the following reformulation of eq. (1):*

$$\min_{f:\mathbb{R}^D \to \mathbb{R}^d} \mathbb{E}_{x \sim f_{\#}\mu} \log \left( 1 + 2\mathbb{E}_{y \sim f_{\#}\mu} \left[ \mathbb{1}_{x \neq y} \exp\left( \mathrm{sim}_f(x, y)/\tau \right) \right] \right)$$

$$= \min_{\rho \in \mathbb{P}(\mathbb{R}^d)} \mathbb{E}_{x \sim \rho} \log \left( 1 + 2\mathbb{E}_{y \sim \rho} \left[ \mathbb{1}_{x \neq y} \exp\left( \mathrm{sim}(x, y)/\tau \right) \right] \right),$$

*where $\mathrm{sim}(x, y) = \mathrm{sim}_{\mathrm{Id}}(x, y) = \frac{x \cdot y}{\|x\|\|y\|}$.*

The result in Proposition B.1 shows that minimizing the NT-Xent cost with respect to an embedding map, once the map is invariant, is equivalent to minimizing over the probability distribution in the latent space. This minimization is *completely independent* of the input data distribution $\mu$.

We further show that two other popular methods for learning dataset invariance with deep learning models exhibit a similar phenomenon as in Proposition B.1, namely, that the loss function itself becomes independent of the original data structure if the embedding map becomes invariant.

First, consider the VICReg Bardes et al. (2021) loss. Given a data distribution $\mu \in \mathbb{P}(\mathbb{R}^D)$ and a distribution for the perturbation functions $\nu$, VICReg minimizes

$$\min_{f:\mathbb{R}^D \to \mathbb{R}^d} \mathbb{E}_{f,g \sim \nu} \mathbb{E}_{x_1, \cdots, x_n \sim \mu} \frac{\lambda_1}{n} \sum_{i=1}^{n} \|f(f(x_i)) - f(g(x_i))\|^2$$

$$+ \lambda_2 \Big( v(f(f(x_1)), \cdots, f(f(x_n))) + v(f(g(x_1)), \cdots, f(g(x_n))) \Big)$$

$$+ \lambda_3 \Big( c(f(f(x_1)), \cdots, f(f(x_n))) + c(f(g(x_1)), \cdots, f(g(x_n))) \Big)$$

where $\lambda_1$, $\lambda_2$, and $\lambda_3$ are hyperparameters. The first term ensures the invariance of $f$ with respect to perturbation functions from $\nu$, $v$ maintains the variance of each embedding dimension, and $c$ regularizes the covariance between pairs of embedded points towards zero. Suppose $f$ is an invariant embedding map such that $f(T(x)) = f(x)$ for all $T \sim \nu$. Then, the above minimization problem

becomes

$$
\min_{f:\mathbb{R}^D\to\mathbb{R}^d}\mathbb{E}_{f,g\sim\nu}\mathbb{E}_{x_1,\cdots,x_n\sim\mu}\lambda_2\Big(v(f(x_1),\cdots,f(x_n))+v(f(x_1),\cdots,f(x_n))\Big)
$$

$$
+\lambda_3\Big(c(f(x_1),\cdots,f(x_n))+c(f(x_1),\cdots,f(x_n))\Big)
$$

$$
=\min_{f:\mathbb{R}^D\to\mathbb{R}^d}\mathbb{E}_{y_1,\cdots,y_n\sim f_\#\mu}\lambda_2\Big(v(y_1,\cdots,y_n)+v(y_1,\cdots,y_n)\Big)
$$

$$
+\lambda_3\Big(c(y_1,\cdots,y_n)+c(y_1,\cdots,y_n)\Big)
$$

Similar to the result in Proposition B.1, the invariance term vanishes. This minimization can now be expressed as a minimization over the embedded distribution:

$$
\min_{\rho\in\mathbb{P}(\mathbb{R}^d)}\mathbb{E}_{y_1,\cdots,y_n\sim\rho}\lambda_2\Big(v(y_1,\cdots,y_n)+v(y_1,\cdots,y_n)\Big)
$$

$$
+\lambda_3\Big(c(y_1,\cdots,y_n)+c(y_1,\cdots,y_n)\Big)
$$

This shows that given an invariant map $f$, the minimization problem becomes completely independent of the input data $\mu$, thus demonstrating the same ill-posedness as the NT-Xent loss in Proposition B.1.

Now, consider the loss function from BYOL Grill et al. (2020). Given a data distribution $\mu\in\mathbb{P}(\mathbb{R}^D)$ and a distribution for the perturbation functions $\nu$, the loss takes the form

$$
\min_{f,q}\mathbb{E}_{f,g\sim\nu}\mathbb{E}_{x\sim\mu}\|q(f(T(x)))-f(T'(x))\|^2
$$

where $q:\mathbb{R}^d\to\mathbb{R}^d$ is an auxiliary function designed to prevent $f$ from collapsing all points $x$ to a constant in $\mathbb{R}^d$. Similar to the previous case, if we assume an invariant map $f$, the above problem becomes

$$
\min_{f,q}\mathbb{E}_{x\sim\mu}\|q(f(x))-f(x)\|^2=\min_{f,q}\mathbb{E}_{y\sim f_\#\mu}\|q(y)-y\|^2
$$

where the second equality follows from a change of variables. Again, this minimization problem can be written with respect to the embedded distribution as:

$$
\min_{\rho\in\mathbb{P}(\mathbb{R}^d),q}\mathbb{E}_{y\sim\rho}\|q(y)-y\|^2
$$

This again shows that once the invariant map is considered, the minimization problem becomes completely independent of the input data $\mu$, highlighting the ill-posedness of the cost function.

## B.2 FURTHER ANALYSIS OF THE STATIONARY POINTS OF NT-XENT LOSS

Our first result provides the first order optimality conditions of the NT-Xent loss eq. (3).

**Proposition B.2.** *The first optimality condition of the problem eq.* (3) *takes the form*

$$
\int_{\mathcal{T}}\int_{\mathbb{R}^D}\Big\langle\int_{\mathcal{T}}\int_{\mathbb{R}^D}\Big(\frac{\Psi'(G_{T,T'}(f,x))}{\eta_f(T(x),T'(x))}+\frac{\Psi'(G_{T,T'}(f,y))}{\eta_f(T(y),T'(y))}\Big)\eta_f'(T(x),T'(y))\big(f(T(x))-f(T'(y))\big)
$$

$$
-\big(\Psi'(G_{T,T'}(f,x))\eta_f(T(x),T'(y))+\Psi'(G_{T',T}(f,x))\eta_f(T'(x),T(y))\big)
$$

$$
\frac{\eta_f'(T(x),T'(x))}{\eta_f^2(T(x),T'(x))}\big(f(T(x))-f(T'(x))\big)d\mu(y)d\nu(T'),h(T(x))\Big\rangle d\mu(x)d\nu(T)=0 \quad (13)
$$

*for all $h$ such that $f+h\in\mathcal{C}$ where $\eta_f'(x,y)=\eta'(\|f(x)-f(y)\|^2/2)$, $\Psi(t)=\log(1+t)$ and $G_{T,T'}(f,x)=\frac{\mathbb{E}_{z\sim\mu}\eta_f(T(x),T'(z))}{\eta_f(T(x),T'(x))}$.*

*If $f$ is invariant to the perturbation in $\nu$, then the gradient of $L$ takes the form*

$$
\nabla L(f)(x)=\int_{\mathbb{R}^D}\big(\Psi'(G_{\mathrm{Id,Id}}(f,x))+\Psi'(G_{\mathrm{Id,Id}}(f,y))\big)\eta_f'(x,y)(f(x)-f(y))d\mu(y). \quad (14)
$$

Using the first optimality condition described in Proposition B.2, we can characterize the minimizer of the NT-Xent loss in eq. (3). The following theorem describes the possible local minimizers of eq. (3), considering the constraint set defined as $\mathcal{C} = \{f : \mathbb{R}^D \to \mathbb{S}^{d-1}\}$.

From the modified formulation eq. (3), we can define a minimizer that minimizes the function $L(f)$ on a constraint set $\mathcal{C} = \{f : \mathbb{R}^D \to \mathbb{R}^d\}$. The following proposition provides insight into the minimizer of eq. (3). The proof is provided in the appendix.

**Proposition B.3.** *This proposition describes three different possible local minimizers of eq. (3) that satisfy the Euler-Lagrange equation in eq. (13).*

1. *Any map $f : \mathbb{R}^D \to \mathbb{R}^d$ that maps to a constant, such that*

$$f(x) = c \in \mathbb{R}^d, \quad \forall x \in \mathcal{M}.$$

2. *In addition to the condition in eq. (4), suppose the attraction and repulsion similarity functions $a : \mathbb{R}_{\geq 0} \to \mathbb{R}$ and $r : \mathbb{R}_{\geq 0} \to \mathbb{R}$ satisfy the following properties:*

   (a) *Each function is maximized at 0, where its value is 1.*
   (b) *Each function satisfies $\lim_{t \to \infty} a(t) = 0$ and $\lim_{t \to \infty} t a'(t) = 0$.*

   *Let $f$ be a map invariant to $\mathcal{T}$. Consider a sequence of maps $\{f_k\}$ such that*

$$f_k(x) = k f(x), \quad \forall x \in \mathcal{M}, \forall k \in \mathbb{N}.$$

   *The limit $f_* = \lim_{k \to \infty} f_k$ satisfies the Euler-Lagrange equation eq. (13).*

*Proof of Proposition B.3.* If $f$ is a constant function, it is trivial that it satisfies eq. (13).

Let us prove the second part of the proposition. From the Euler-Lagrange equation in eq. (13), by plugging in $f_k$ and using the fact that $f$ is invariant to $\mathcal{T}$, the Euler-Lagrange equation can be simplified to

$$\int_{\mathbb{R}^D} \left( \Psi'(G(f_k, \mathrm{Id}, x)) + \Psi'(G(f_k, \mathrm{Id}, y)) \right) r'_{f_k}(x, y) \langle f_k(x) - f_k(y), h(x) \rangle \, d\mu(y)$$

for any $h : \mathbb{R}^D \to \mathbb{R}^d$. Using the invariance of $f_k$, we have

$$= \int_{\mathbb{R}^D} \left( \Psi'(G(f_k, \mathrm{Id}, x)) + \Psi'(G(f_k, \mathrm{Id}, y)) \right) \left( k r'_{f_k}(x, y) \right) \langle f(x) - f(y), h(x) \rangle \, d\mu(y). \tag{15}$$

Furthermore, by the assumptions on the function $r$,

$$k r'_{f_k}(x, y) = k r' \left( \frac{k^2 \|f(x) - f(y)\|^2}{2} \right) \to 0, \quad \text{as } k \to \infty$$

$$\Psi'(G(f_k, \mathrm{Id}, x)) = \Psi' \left( \mathbb{E}_{z \sim \mu} r \left( \frac{k^2 \|f(x) - f(z)\|^2}{2} \right) \right) \to \Psi'(0), \quad \text{as } k \to \infty.$$

Thus, eq. (15) converges to 0 as $k \to \infty$. This proves the theorem. $\square$

### B.3 PROOF OF THEOREM 3.1

First, we prove Theorem 3.1, which characterizes the stationary points of the loss function. After the proof, we demonstrate that by considering an additional condition on the direction of the second variation at the stationary points, it is the second variation is strictly positive, thereby showing that the stationary point is a local minimizer under this condition.

*Proof of Theorem 3.1.* We consider the following problem:

$$\min_{f : \mathbb{R}^D \to \mathbb{S}^{d-1}} L(f), \tag{16}$$

where $L$ is a loss function defined in eq. (3). The problem in eq. (16) can be reformulated as a constrained minimization problem:

$$\min_{\substack{f : \mathbb{R}^D \to \mathbb{R}^d \\ \|f\| = 1}} L(f).$$

By relaxing the the constraint for $\|f\| = 1$, we can derive the lower bound such that

$$\min_{\substack{f:\mathbb{R}^D \to \mathbb{R}^d \\ \|f\|=1}} L(f) \geq \min_{\substack{f:\mathbb{R}^D \to \mathbb{R}^d \\ \int_{\mathbb{R}^D} \|f\| d\mu = 1}} L(f).$$

Note that since the constraint sets satisfy $\{\|f\| = 1\} \subset \{\int_{\mathbb{R}^D} \|f\| d\mu = 1\}$, the stationary point from from the latter constraint set is also the stationary point of the prior set.

By introducing the Lagrange multiplier $\lambda$ for the constraint, we can convert the minimization problem into a minimax problem:

$$\min_{f:\mathbb{R}^D \to \mathbb{R}^d} \max_{\lambda \in \mathbb{R}} \left[ L(f) + \lambda \mathbb{E}_{x \sim \mu}(1 - \|f(x)\|) \right]. \tag{17}$$

Using the Euler-Lagrange formulation in eq. (13), we can derive the Euler-Lagrange equation for the above problem, incorporating the Lagrange multiplier. To show that $f$ is a minimizer of the problem in eq. (17), we need to demonstrate that there exists $\lambda \in \mathbb{R}$ such that the following equation holds:

$$\int_{\mathbb{R}^D} \left[ \Psi'(G(f,x)) + \Psi'(G(f,y)) \right] \eta_f'(x,y)(f(x) - f(y)) \, d\mu(y) - \lambda \frac{f(x)}{\|f(x)\|} = 0,$$

for all $x \in \mathcal{M}$. Note that since $f$ is an invariant map, $f$ disappears and $\eta_f(x, f(x)) = 1$. Furthermore, since $f$ maps onto $\mathbb{S}^{d-1}$, we have $\|f(x)\| = 1$ for all $x \in \mathbb{R}^D$. Additionally, using the change of variables, we obtain

$$\lambda = C \int_{\mathbb{S}^{d-1}} r'(|x - y|^2/2)(x - y) \, df_{\#}\mu(y), \tag{18}$$

where $C$ is defined as $C = \Psi'(\mathbb{E}_{z \sim f_{\#}\mu} \left[ r(|x_0 - z|^2/2) \right])$ for $x_0 \sim f_{\#}\mu$. Given that the function

$$h(x, y) = r'(|x - y|^2/2)(x - y)$$

is an anti-symmetric function, by the assumption on $f_{\#}\mu$ in eq. (5), the integral on the right-hand side of eq. (18) is constant for all $x \sim f_{\#}\mu$. Therefore, by defining $\lambda$ as in eq. (18), this proves the lemma. $\qquad \square$

Now that we have identified the characteristics required for embedding maps to be stationary points, the next lemma shows that the second variation at this stationary point, in a specific direction $h$, is positive. This demonstrates that the stationary point is indeed a local minimizer along this particular direction.

**Lemma B.4.** *Fix $\tau > 0$ and define $\eta_f(x, y) = e^{-\|f(x)-f(y)\|^2/2\tau}$. Let $f : \mathbb{R}^D \to \mathbb{S}^{d-1}$ be an embedding map such that the embedded distribution $f_{\#}\mu = \sum_{i=1}^n \delta_{x_i}$ is a discrete measure on $\mathbb{S}^{d-1}$, satisfying that the number of points $n = Km$, where $K$ is the number of cluster centers $\{\xi_i\}_{i=1}^K$ and $m$ is some positive integer. Moreover, the points satisfy the condition:*

$$x_i = \xi_{\lfloor i/K \rfloor + 1} \quad for \ i \in [\![n]\!]. \tag{19}$$

*Furthermore, let $\sigma > 0$ be a positive constant satisfying $\sigma > 3K^2\tau$. Then,*

$$\delta^2 L(f)(h, h) > 0$$

*for any $h : \mathbb{R}^D \to \mathbb{R}^d$ satisfying $f + h \in \mathbb{S}^{d-1}$ and*

$$\left( \langle f(\xi_i) - f(\xi_j), h(\xi_i) - h(\xi_j) \rangle \right)^2 \geq \sigma \|h(\xi_i) - h(\xi_j)\|^2. \tag{20}$$

*Proof.* Let $f : \mathbb{R}^D \to \mathbb{R}^d$ be an invariant embedding map. From Proposition B.2, the first variation takes the form

$$\int_{\mathbb{R}^D} \Psi'(G(f,x)) \int_{\mathbb{R}^D} \eta_f'(x,y) \langle f(x) - f(y), h(x) - h(y) \rangle d\mu(y) d\mu(x).$$

The second variation takes the form

$$\int_{\mathbb{R}^D} \Psi''(G(f,x)) \left( \int_{\mathbb{R}^D} \eta_f'(x,y)\langle f(x)-f(y), h(x)-h(y)\rangle d\mu(y)\right)^2 d\mu(x)$$

$$+\int_{\mathbb{R}^D} \Psi'(G(f,x)) \int_{\mathbb{R}^D} r_f''(x,y)\Big(\langle f(x)-f(y), h(x)-h(y)\rangle\Big)^2 d\mu(y)d\mu(x)$$

$$+\int_{\mathbb{R}^D} \Psi'(G(f,x)) \int_{\mathbb{R}^D} \eta_f'(x,y)\|h(x)-h(y)\|^2 d\mu(y)d\mu(x).$$

For simplicity, let us choose explicit forms for $\Psi$ and $r$. The proof will be general enough to apply to any $\Psi$ and $r$ that satisfy the conditions mentioned in the paper. Let $\Psi(t) = \log(1 + t/2)$ and $r(t) = e^{-t/(2\tau)}$. With these choice of functions and by the change of variables,

$$=-\frac{1}{\tau^2}\int_{\mathbb{S}^{d-1}}\left(\frac{1}{1+G(x)^2/2}\right)^2\left(\int_{\mathbb{S}^{d-1}}e^{-\|x-y\|^2/(2\tau)}\langle x-y, T'(x)-T'(y)\rangle df_{\#}\mu(y)\right)^2 df_{\#}\mu(x)$$

$$+\frac{1}{\tau^2}\int_{\mathbb{S}^{d-1}}\frac{1}{1+G(x)^2/2}\int_{\mathbb{S}^{d-1}}e^{-\|x-y\|^2/(2\tau)}\Big(\langle x-y, T'(x)-T'(y)\rangle\Big)^2 df_{\#}\mu(y)df_{\#}\mu(x)$$

$$-\frac{1}{\tau}\int_{\mathbb{S}^{d-1}}\frac{1}{1+G(x)^2/2}\int_{\mathbb{S}^{d-1}}e^{-\|x-y\|^2/(2\tau)}\|T'(x)-T'(y)\|^2 df_{\#}\mu(y)df_{\#}\mu(x). \tag{21}$$

where $G(x) = \mathbb{E}_{y\sim f_{\#}\mu}e^{-\|x-y\|^2/(2\tau)}$ and $T'(x) = h(f^{-1}(x))$. By Jensen's inequality, we have

$$\left(\int_{\mathbb{S}^{d-1}}e^{-\|x-y\|^2/(2\tau)}\langle x-y, T'(x)-T'(y)\rangle df_{\#}\mu(y)\right)^2$$

$$\leq \int_{\mathbb{S}^{d-1}}e^{-\|x-y\|^2/(2\tau)}\Big(\langle x-y, T'(x)-T'(y)\rangle\Big)^2 df_{\#}\mu(y).$$

Therefore, eq. (21) can be bounded below by

$$\geq\frac{1}{\tau^2}\int_{\mathbb{S}^{d-1}}\frac{G(x)^2/2}{(1+G(x)^2/2)^2}\int_{\mathbb{S}^{d-1}}e^{-\|x-y\|^2/(2\tau)}\Big(\langle x-y, T'(x)-T'(y)\rangle\Big)^2 df_{\#}\mu(y)df_{\#}\mu(x)$$

$$-\frac{1}{\tau}\int_{\mathbb{S}^{d-1}}\frac{1}{1+G(x)^2/2}\int_{\mathbb{S}^{d-1}}e^{-\|x-y\|^2/(2\tau)}\|T'(x)-T'(y)\|^2 df_{\#}\mu(y)df_{\#}\mu(x)$$

$$=\frac{1}{\tau^2}\int_{\mathbb{S}^{d-1}}\frac{1}{1+G(x)^2/2}\int_{\mathbb{S}^{d-1}}e^{-\|x-y\|^2/(2\tau)}$$

$$\left(\frac{G(x)^2}{2(1+G(x)^2/2)}\Big(\langle x-y, T'(x)-T'(y)\rangle\Big)^2 - \tau\|T'(x)-T'(y)\|^2\right)df_{\#}\mu(y)df_{\#}\mu(x).$$

By the assumption on $f_{\#}\mu$ in eq. (19), the above can be written as

$$=\frac{1}{n^2\tau^2}\sum_{i=1}^{n}\frac{1}{1+\tilde{G}(x_i)^2/2}\sum_{\substack{j=1\\j\neq i}}^{n}e^{-\|x_i-x_j\|^2/(2\tau)}$$

$$\left(\frac{\tilde{G}(x_i)^2}{2(1+\tilde{G}(x)^2/2)}\Big(\langle x_i-x_j, g(x_i)-g(x_j)\rangle\Big)^2 - \tau\|g(x_i)-g(x_j)\|^2\right)$$

$$=\frac{m^2}{n^2\tau^2}\sum_{i=1}^{K}\frac{1}{1+\tilde{G}(\xi_i)^2/2}\sum_{\substack{j=1\\j\neq i}}^{K}e^{-\|\xi_i-\xi_j\|^2/(2\tau)}$$

$$\left(\frac{\tilde{G}(\xi_i)^2}{2(1+\tilde{G}(\xi_i)^2/2)}\Big(\langle \xi_i-\xi_j, g(\xi_i)-g(\xi_j)\rangle\Big)^2 - \tau\|g(\xi_i)-g(\xi_j)\|^2\right) \tag{22}$$

If $K = 1$, the second variation becomes 0, and is therefore nonnegative. Now, suppose $K > 1$. We can bound $\tilde{G}$ from below by

$$\tilde{G}(\xi_i) = \frac{1}{K}\sum_{k=1}^{K}e^{-\|\xi_i-\xi_k\|^2/(2\tau)} \geq \frac{1}{K}. \tag{23}$$

Furthermore, from the condition in eq. (20), we have

$$\Big(\langle \xi_i - \xi_j, g(\xi_i) - g(\xi_j)\rangle\Big)^2 \geq \sigma \|g(\xi_i) - g(\xi_j)\|^2 \tag{24}$$

for some positive constant $\sigma > 0$. Combining eq. (23) and eq. (24), we can bound eq. (22) from below by

$$\geq \frac{1}{K^2 \tau^2} \sum_{i=1}^{K} \frac{1}{1 + \tilde{G}(\xi_i)^2/2} \sum_{\substack{j=1 \\ j \neq i}}^{K} e^{-\|\xi_i - \xi_j\|^2/(2\tau)} \Big(\frac{\sigma}{3K^2} - \tau\Big) \|g(\xi_i) - g(\xi_j)\|^2.$$

By the condition on $\sigma$, the above quantity is strictly greater than zero. This concludes the proof of the lemma.

$\square$

### B.4 PROOF OF PROPOSITION 4.1

*Proof.* The gradient of $\mathcal{L}$ is given by

$$\nabla \mathcal{L}(w) = \frac{1}{n} \sum_{i=1}^{n} \sum_{k=1}^{d} \nabla_{z^k} L(f(w, x_i), x_i) \nabla_w f^k(w, x_i) \tag{25}$$

where $\nabla_{z^k} L(f(w, x_i))$ is a gradient of $L$ with respect to $k$-th coordinate.

For simplicity of notation, let us denote by

$$f_i^k = f^k(w, x_i), \quad L_i = L(f(w, x_i), x_i).$$

Thus, eq. (25) can be rewritten as

$$\nabla \mathcal{L}(w) = \frac{1}{n} \sum_{i=1}^{n} \sum_{k=1}^{d} \nabla_{z^k} L_i \nabla_w f_i^k \tag{26}$$

By the definition of the loss function in eq. (6), $w(t)$ satisfies the gradient flow such that

$$\dot{w}(t) = -\nabla \mathcal{L}(w). \tag{27}$$

Thus, the solution of the above ODE converges to the local minimizer of $\mathcal{L}$ as $t$ grows.

For each $i \in [\![n]\!]$ and $k \in [\![d]\!]$, denote by

$$z_i^k(t) = f_i^k(t).$$

Let us compute the time derivative of $z_i^k$. Using a chain rule, eq. (26) and eq. (27),

$$\dot{z}_i^k(t) = \nabla_w f_i^k \cdot \dot{w}(t) = -\nabla_w f_i^k \cdot \nabla \mathcal{L}(w) = -\frac{1}{n} \sum_{j=1}^{m} \sum_{l=1}^{d} \nabla_w f_i^k \cdot \nabla_w f_j^l \nabla_{y^l} L_j. \tag{28}$$

Using eq. (8), eq. (28), $\dot{z}_i(t)$ can be written as

$$\dot{z}_i(t) = -\frac{1}{n} \sum_{j=1}^{n} K_{ij} \nabla L_j(t).$$

This completes the proof. $\square$

### B.5 PROOF OF THEOREM 4.2

*Proof.* Consider the gradient descent iterations in eq. (10). Suppose $f$ is invariant to the perturbation from $\nu$ at $b$-th iteration, that is we have $f(w^{(b)}, f(x)) = f(w^{(b)}, x)$ for all $x \sim \mu$ and $T \sim \nu$. We

want to show that, given an invariant embedding map $f(w^{(b)}, \cdot)$, it remains invariant after iteration $b$. From the gradient formulation of the loss function in Proposition B.2, we have

$$\nabla L(f(w^{(b)}, x), x) = -\int_{\mathbb{R}^D} \Psi'(x, y)(f(w^{(b)}, x) - f(w^{(b)}, y))\, d\mu(y),$$

where $\Psi'(x, y) = \Big(\Psi'(G(f(w^{(b)}, \cdot), x)) + \Psi'(G(f(w^{(b)}, \cdot), y))\Big)\eta'_{f(w^{(b)}, \cdot)}(x, y)$.

From the gradient descent formulation in eq. (10), we have

$$f(w^{(b+1)}, f(x_i)) = f(w^{(b)}, f(x_i)) - \sigma \nabla L(f(w^{(b)}, f(x_i))),$$

which gives

$$f(w^{(b+1)}, f(x_i)) = f(w^{(b)}, f(x_i)) + \sigma \int_{\mathbb{R}^D} \Psi'(x, y)(f(w^{(b)}, f(x_i)) - f(w^{(b)}, y))\, d\mu(y),$$

and since $f(w^{(b)}, f(x_i)) = f(w^{(b)}, x_i)$ by invariance, this simplifies to

$$f(w^{(b+1)}, x_i) - \sigma \nabla L(f(w^{(b)}, x_i), x_i) = f(w^{(b+1)}, x_i),$$

which shows that $f(w^{(b+1)}, x_i)$ is invariant for all $i \in [\![n]\!]$. Therefore, the embedding map remains invariant throughout the optimization process.

Now, consider the gradient descent iteration with a neural network in eq. (11). Suppose $f$ is invariant to perturbations from $\nu$ and satisfies

$$\nabla_w f(w^{(b)}, f(x)) = \nabla_w f(w^{(b)}, x), \quad \forall x \sim \mu, f \sim \nu. \tag{29}$$

Denote the kernel matrix function $K_{ij}$ given a perturbation function $T \sim \nu$ as

$$(K_{ij}(w^{(b)}, f))^{kl} = (\nabla_w f^k(w^{(b)}, f(x_i)))^\top (\nabla_w f^l(w^{(b)}, f(x_j))).$$

Then, we have

$$f(w^{(b+1)}, f(x_i)) = f(w^{(b)}, f(x_i)) - \frac{\sigma}{n} \sum_{j=1}^n K_{ij}(w^{(b)}, f)\nabla L(f(w^{(b)}, f(x_i)), x_i)$$

$$= f(w^{(b)}, x_i) - \frac{\sigma}{n} \sum_{j=1}^n K_{ij}(w^{(b)}, \mathrm{Id})\nabla L(f(w^{(b)}, x_i), x_i)$$

$$= f(w^{(b+1)}, x_i).$$

Thus, if $f$ is invariant at the $b$-th iteration, it remains invariant. However, note that this result no longer holds if the condition in eq. (29) fails, meaning that $f$ is not invariant for $b + 1$-th iteration. This completes the proof. $\qquad\square$

## B.6 Explicit Formula for the Neural Network Kernel Matrix

In this section, we derive an explicit formula for the kernel matrix to better understand how the neural network kernel influences the gradient descent dynamics of $f(w(t), x)$ for data points $x \in X$. This formula provides insight into how the kernel governs the evolution of features during training.

While the main paper focuses on a simplified case where the embedding map outputs real-valued features (i.e., $f : \mathbb{R}^{MD+D} \to \mathbb{R}$), here we consider a more general setting where the embedding map outputs vector-valued features in $\mathbb{R}^d$. This generalization allows us to derive a broader formula for the kernel matrix. Specifically, we consider a one-hidden-layer fully connected neural network of the form $f : \mathbb{R}^{MDd+D} \to \mathbb{R}^d$.

$$f(w(t), x) = A^\top \sigma(w(t)x), \tag{30}$$

where $x \in \mathbb{R}^D$ is a data sample and $A \in \mathbb{R}^{Md \times d}$ is a constant matrix defined as

$$A = \frac{1}{\sqrt{MD}} \begin{bmatrix} \mathbb{1}_{M\times 1} & \mathbb{0}_{M\times 1} & \cdots & \mathbb{0}_{M\times 1} \\ \mathbb{0}_{M\times 1} & \mathbb{1}_{M\times 1} & \cdots & \mathbb{0}_{M\times 1} \\ \vdots & \vdots & \ddots & \vdots \\ \mathbb{0}_{M\times 1} & \mathbb{0}_{M\times 1} & \cdots & \mathbb{1}_{M\times 1} \end{bmatrix} \in \mathbb{R}^{Md \times d}, \tag{31}$$

where $\mathbb{1}_{M \times 1}$ and $\mathbb{0}_{M \times 1}$ represent the $M$-dimensional vectors of ones and zeros, respectively. Additionally, $w(t) = (b_p^k(t))_{k \in [\![D]\!], p \in [\![Md]\!]} \in \mathbb{R}^{Md \times D}$ is the weight matrix, and $\sigma$ is a differentiable activation function applied element-wise.

Note that $A$ acts as an averaging matrix that, when multiplied by the $(Md)$-dimensional vector $\sigma(w(t)x)$, produces a $d$-dimensional vector. Furthermore, we assume that the parameters of $w(t)$ are uniformly bounded, such that there exists a constant $C$ with $|b_p^k(t)| < C$ for all $t \geq 0$, $k$, and $p$.

**Proposition B.5.** *Given the description of the embedding map in eq. (30), the kernel matrix $K_{ij}^{kl}$ defined in eq. (8) can be explicitly written as*

$$K_{ij}^{kl} = \frac{\mathbb{1}_{k=l}}{MD} x_i^\top x_j \sum_{p=(k-1)M+1}^{kM} \sigma'(b_p x_i)\sigma'(b_p x_j). \tag{32}$$

*where $\mathbb{1}_{k=l}$ is an indicator function that equals $1$ if $k = l$ and $0$ otherwise.*

*Proof.* We describe the matrices $A \in \mathbb{R}^{Md \times d}$ and $B \in \mathbb{R}^{Md \times D}$ as follows:

$$A = \begin{bmatrix} | & | & \cdots & | \\ a^1 & a^2 & \cdots & a^d \\ | & | & \cdots & | \end{bmatrix} = \begin{bmatrix} - & a_1 & - \\ & \vdots & \\ - & a_{Md} & - \end{bmatrix} = \begin{bmatrix} a_1^1 & \cdots & a_1^d \\ \vdots & \ddots & \vdots \\ a_{Md}^1 & \cdots & a_{Md}^d \end{bmatrix},$$

$$B(t) = \begin{bmatrix} | & | & \cdots & | \\ b^1(t) & b^2(t) & \cdots & b^D(t) \\ | & | & \cdots & | \end{bmatrix} = \begin{bmatrix} - & b_1(t) & - \\ & \vdots & \\ - & b_{Md}(t) & - \end{bmatrix} = \begin{bmatrix} b_1^1(t) & \cdots & b_1^D(t) \\ \vdots & \ddots & \vdots \\ b_{Md}^1(t) & \cdots & b_{Md}^D(t) \end{bmatrix}.$$

In this notation, $a^k$ and $b^k$ are $Md$-dimensional column vectors, $a_p$ and $b_p$ are $d$- and $D$-dimensional row vectors, and $a_p^k$ and $b_p^k$ are scalars.

We can write $f^k$ with respect to $a_i^k$ and $b_i^k$.

$$f^k(B, x) = (a^k)^\top \sigma(Bx) = \sum_{i=1}^{Md} a_i^k \sigma\left(b_i x\right) = \frac{1}{\sqrt{MD}} \sum_{i=(k-1)M+1}^{kM} \sigma\left(b_i x\right)$$

where the last equality uses the definition of a matrix $A$ in eq. (31). By differentiating with respect to $b_i^l$, we can derive explicit forms for the gradient of $f^k$ with respect to a weight matrix $B$.

$$\nabla_w f^k(B, x) = \left(a^k \odot \sigma'(Bx)\right)x^\top$$

$$= \begin{bmatrix} a_1^k \sigma'(b_1 x)x^1 & \cdots & a_1^k \sigma'(b_1 x)x^D \\ \vdots & \ddots & \vdots \\ a_M^k \sigma'(b_M x)x^1 & \cdots & a_M^k \sigma'(b_M x)x^D \end{bmatrix}$$

$$= \frac{1}{\sqrt{MD}} \begin{bmatrix} 0 & \cdots & 0 \\ \vdots & \ddots & \vdots \\ 0 & \cdots & 0 \\ \sigma'(b_1 x)x^1 & \cdots & \sigma'(b_1 x)x^D \\ \vdots & \ddots & \vdots \\ \sigma'(b_M x)x^1 & \cdots & \sigma'(b_M x)x^D \\ 0 & \cdots & 0 \\ \vdots & \ddots & \vdots \\ 0 & \cdots & 0 \end{bmatrix} \in \mathbb{R}^{Md \times D}$$

where the row index of nonzero entries ranges from $(k-1)M + 1$ to $kM$.

Define an inner product such that for $h \in \mathbb{R}^{Md \times D}$

$$\langle \nabla_w f^k(B, x), h \rangle, \quad k \in [\![D]\!].$$

Now we are ready to show the explicit formula of the inner product $\langle \nabla_w f^k, \nabla_w f^l \rangle$.

$$\langle \nabla_w f^k(B, x_i), \nabla_w f^l(B, x_j) \rangle = \frac{\mathbb{1}_{k=l}}{MD}(x_i^\top x_j) \sum_{p=(k-1)M+1}^{kM} \sigma'(b_p x_i)\sigma'(b_p x_j)$$

where $\mathbb{1}_{k=l}$ is an indicator function that equals 1 if $k = l$ and 0 otherwise. Therefore, the kernel matrix takes the form

$$(K^{kl})_{ij} = \frac{\mathbb{1}_{k=l}}{MD}(x_i^\top x_j) \sum_{p=(k-1)M+1}^{kM} \sigma'(b_p x_i)\sigma'(b_p x_j).$$

$\square$

From Proposition B.5, as done in NTK paper (Jacot et al., 2018), one can consider how the kernel converges as the width of the neural network approaches infinity, i.e., as $M \to \infty$ in eq. (30). The following proposition shows the formulation of the limiting kernel in the infinite-width neural network.

**Proposition B.6.** *Suppose the weight matrix $B$ satisfies that each row vector $b_i$, for $i \in \{1, \ldots, Md\}$, consists of independent and identically distributed random variables in $\mathbb{R}^D$ with a Gaussian distribution. Also, suppose the activation function is $\sigma(x) = x_+ = \max\{x, 0\}$. Then, as $M \to \infty$, the kernel matrix converges to $K^\infty \in \mathbb{R}^{d \times d}$, where*

$$K_{ij}^\infty = \frac{x_i^\top x_j}{D}\left[\frac{1}{2} - \frac{1}{2\pi}\arccos\left(\frac{x_i^\top x_j}{\|x_i\|\|x_j\|}\right)\right] \boldsymbol{I}_{d \times d}, \quad \boldsymbol{I}_{d \times d} \text{ is an identity matrix.}$$

## C    PROOF OF THEOREM 4.3

*Proof of Theorem 4.3.* **Step 1.** Using (a) and (b), expand the gradient of equation 12 with respect to $z_i$ and use $\exp(s) = 1 + \mathcal{O}(s)$ for small $s$:

$$\nabla L(z_i, x_i) = -\frac{1}{n\tau} \mathbb{E}_T \sum_{j=1}^n e^{-\frac{\|z_i - f(w, T(x_j))\|^2}{2\tau}} \left(z_i - f(w, T(x_j))\right)$$

$$+ \frac{2}{n\tau} \sum_{j \in X_{\Gamma(i)}} (z_i - z_j) + \mathcal{O}\left(\frac{\varepsilon^3}{\tau^2} + \frac{\gamma}{\tau}\right)$$

$$\approx -\frac{1}{n\tau} \mathbb{E}_T \sum_{j=1}^n \left(z_i - f(w, T(x_j))\right) + \frac{2}{n\tau} \sum_{j \in X_{\Gamma(i)}} (z_i - z_j) + \mathcal{O}\left(\frac{\varepsilon^3}{\tau^2} + \frac{\gamma}{\tau}\right)$$

$$= \frac{1}{n\tau}\left(\sum_{j=1}^n z_j - 2\sum_{j \in X_{\Gamma(i)}} z_j\right) + \mathcal{O}\left(\frac{\varepsilon^3}{\tau^2} + \frac{\gamma}{\tau}\right).$$

Stacking over $i$ gives

$$\nabla L = \frac{1}{\tau}(J_1 - 2J_2)z + \mathcal{O}\left(\frac{\varepsilon^3}{\tau^2} + \frac{\gamma}{\tau}\right),$$

where $J_1 \in \mathbb{R}^{n \times n}$ has all entries $1/n$, and $J_2$ is block diagonal with equal $1/n$ entries within each cluster block.

**Step 2.** Let $K(0)$ denote the NTK Gram matrix at $t = 0$, with entries $\langle g_i, g_j \rangle$. Writing $g_i = \mu_q + \xi_i$ for $i \in X_q$ and using $\sum_{i \in X_q} \xi_i = 0$ together with $\frac{2}{n}\sum_{i \in X_q}\|\xi_i\|^2 \leq \sigma^2$, we obtain

$$K(0) = \begin{bmatrix} a\,\mathbf{1}_{n/2 \times n/2} & b\,\mathbf{1}_{n/2 \times n/2} \\ b\,\mathbf{1}_{n/2 \times n/2} & c\,\mathbf{1}_{n/2 \times n/2} \end{bmatrix} + \mathcal{O}(\sigma^2), \quad a := \|\mu_1\|^2, \ c := \|\mu_2\|^2, \ b := \langle \mu_1, \mu_2 \rangle,$$

so that $a + c - 2b = \|\mu_1 - \mu_2\|^2 = \Theta > 0$.

**Step 3.** The gradient flow is

$$\dot{z}(t) = -\frac{1}{n}K(t)\nabla L = -\frac{1}{n}K(0)\nabla L + R_K(t), \qquad \|R_K(t)\| \le \frac{\delta}{n}\|\nabla L\|.$$

Using Step 1,

$$\dot{z}(t) = -\frac{1}{n\tau}K(0)(J_1 - 2J_2)z + R(t),$$

and, since $(J_1 - 2J_2)z$ is block-constant with $\|(J_1 - 2J_2)z\|_2 = \mathcal{O}(\sqrt{n})$ on $[0, T]$,

$$\|R(t)\| \lesssim \frac{\sigma^2}{\sqrt{n}\,\tau} + \frac{\delta}{\sqrt{n}\,\tau} + \frac{\varepsilon^3}{n\tau^2} + \frac{\gamma}{n\tau}.$$

Abbreviate

$$\tilde{K} := \frac{1}{n\tau}K(0)(J_1 - 2J_2), \qquad \tilde{\sigma}_{\mathrm{rem}} := \frac{\sigma^2}{\sqrt{n}\,\tau} + \frac{\delta}{\sqrt{n}\,\tau} + \frac{\varepsilon^3}{n\tau^2} + \frac{\gamma}{n\tau},$$

so that $R(t) = \mathcal{O}(\tilde{\sigma}_{\mathrm{rem}})$ and

$$\dot{z}(t) = -\tilde{K}z + R(t). \tag{33}$$

With

$$u = \frac{1}{\sqrt{n\xi}}\begin{bmatrix}(a-b)\,\mathbf{1}_{n/2}\\(b-c)\,\mathbf{1}_{n/2}\end{bmatrix}, \qquad \xi := \tfrac{1}{2}\big((a-b)^2 + (b-c)^2\big),$$

we have

$$\tilde{K}u = -\frac{a+c-2b}{4\tau}u + \mathcal{O}\Big(\frac{\sigma^2}{n\tau}\Big) = -\frac{\Theta}{4\tau}u + \mathcal{O}\Big(\frac{\sigma^2}{n\tau}\Big),$$

so, by Weyl's theorem and assumption (e), the eigenvalue is

$$\lambda = -\frac{\Theta}{4\tau} + \mathcal{O}\Big(\frac{\sigma^2}{n\tau}\Big) < 0. \tag{34}$$

**Step 4.** Decompose

$$z(t) = \alpha(t)\,u + z_\perp(t), \qquad \alpha(t) := \langle z(t), u\rangle, \qquad \langle z_\perp(t), u\rangle = 0.$$

From equation 33 and $|\langle R(t), u\rangle| \le \|R(t)\|$,

$$\dot{\alpha}(t) = -\lambda\,\alpha(t) + r_\|(t), \qquad |r_\|(t)| \le C\,\tilde{\sigma}_{\mathrm{rem}}.$$

Solving the linear ODE gives

$$\alpha(t) = \alpha(0)\,e^{-\lambda t} + \mathcal{O}\Big(\frac{\tilde{\sigma}_{\mathrm{rem}}}{-\lambda}\big(e^{-\lambda t} - 1\big)\Big). \tag{35}$$

For the orthogonal component,

$$\|z_\perp(t)\| \le \|z_\perp(0)\| + C\,t\,\tilde{\sigma}_{\mathrm{rem}}. \tag{36}$$

The initial coefficient satisfies

$$\alpha(0) = \langle z(0), u\rangle = \frac{\sqrt{n}}{2\sqrt{\xi}}\Big((a-b)\,\bar{z}^{(1)} + (b-c)\,\bar{z}^{(2)}\Big), \tag{37}$$

and after recentering the global mean, (a) yields

$$|\alpha(0)| = \mathcal{O}(\sqrt{n}\,\varepsilon), \qquad \|z_\perp(0)\| = \mathcal{O}(\sqrt{n}\,\varepsilon).$$

**Step 5.** For $i \in X_1$ and $j \in X_2$,

$$u_i - u_j = \frac{(a-b) - (b-c)}{\sqrt{n\xi}} = \frac{\Theta}{\sqrt{n\xi}},$$

so

$$z_i(t) - z_j(t) = \frac{\Theta}{\sqrt{n\xi}}\,\alpha(t) + \big(z_{\perp,i}(t) - z_{\perp,j}(t)\big). \tag{38}$$

Using equation 36,

$$\left|z_{\perp,i}(t) - z_{\perp,j}(t)\right| \le 2\|z_\perp(t)\| \le 2\|z_\perp(0)\| + C\,t\,\tilde\sigma_{\text{rem}} = \mathcal{O}(\sqrt{n}\,\varepsilon) + C\,t\,\tilde\sigma_{\text{rem}}.$$

Insert equation 35 into equation 38 and choose the sign of $u$ so that $\alpha(0) \ge 0$:

$$z_i(t) - z_j(t) \;\ge\; \frac{\Theta}{\sqrt{n\xi}}\Big(\alpha(0)\,e^{-\lambda t} - C\frac{\tilde\sigma_{\text{rem}}}{-\lambda}\big(e^{-\lambda t} - 1\big)\Big) - \mathcal{O}(\sqrt{n}\,\varepsilon) - C\,t\,\tilde\sigma_{\text{rem}}.$$

With eq. (34) and $|\alpha(0)| = \mathcal{O}(\sqrt{n}\,\varepsilon)$,

$$z_i(t) - z_j(t) \;\ge\; \frac{C\Theta}{\sqrt{\xi}}\,\varepsilon\,e^{-\lambda t} - C\Big(\frac{\tau}{\sqrt{n\xi}}\,\tilde\sigma_{\text{rem}}\,e^{-\lambda t} + \sqrt{n}\,\varepsilon + t\,\tilde\sigma_{\text{rem}}\Big).$$

$$z_i(t) - z_j(t) \;\ge\; C\left(\frac{\Theta\varepsilon}{\sqrt{\xi}} - \frac{\tau\tilde\sigma_{\text{rem}}}{\sqrt{n\xi}}\right)e^{-\lambda t} - C\Big(\sqrt{n}\,\varepsilon + t\,\tilde\sigma_{\text{rem}}\Big). \tag{39}$$

Hence there exists $t_0 \in (0, T]$ with $z_i(t_0) > z_j(t_0)$ provided the leading signal dominates. Assumption (e) is a clean sufficient condition; it is slightly stronger than necessary because the terms multiplied by $\tilde\sigma_{\text{rem}}$ carry extra $1/\sqrt{n}$ or $1/n$ factors. The domination holds uniformly over cross-cluster pairs, so linear separability follows at $t_0$.

Define

$$F(t) \;:=\; \Big(\tfrac{\Theta}{\sqrt{\xi}}\varepsilon - \tfrac{\tau}{\sqrt{n\xi}}\tilde\sigma_{\text{rem}}\Big)e^{-\lambda t} \;-\; C\sqrt{n}\,\varepsilon \;-\; Ct\,\tilde\sigma_{\text{rem}}.$$

From equation 39, we have $z_i(t) - z_j(t) \ge F(t)$. Since $\lambda < 0$, under the condition from assumption (e),

$$\Theta\,\sqrt{n}\,\varepsilon \;\gg\; \frac{\varepsilon^3}{\tau} + \gamma + \sigma^2 + \delta\,,$$

we get $\Theta\sqrt{n}\,\varepsilon \gg \tau\,\tilde\sigma_{\text{rem}}$, hence

$$A := \tfrac{\Theta}{\sqrt{\xi}}\varepsilon \;>\; B := \tfrac{\tau}{\sqrt{n\xi}}\tilde\sigma_{\text{rem}}.$$

Furthermore, $F$ has strictly positive initial slope:

$$F'(0) = -\lambda(A - B) - C\tilde\sigma_{\text{rem}} \;\gtrsim\; \frac{\Theta}{\tau} \cdot \frac{\Theta}{\sqrt{\xi}}\varepsilon - C'\tilde\sigma_{\text{rem}} \;>\; 0,$$

and, since $e^{-\lambda t}$ grows while the error part is at most linear in $t$, the exponential term eventually dominates the constant and linear terms. By continuity, there exists $t_0 \in (0, T]$ with $F(t_0) > 0$, hence

$$z_i(t_0) - z_j(t_0) \;\ge\; F(t_0) \;>\; 0$$

for every $i \in X_1$, $j \in X_2$. This proves linear separability at some time $t_0$ within the window. $\qquad\square$

# D   EXTRA NUMERICAL RESULTS

In this section, we extend the experiment to higher-dimensional settings, where the feature distribution lies in $\mathbb{R}^2$ and $\mathbb{R}^3$ instead of $\mathbb{R}$, and present additional results to support Theorem 4.3. Furthermore, we constrain the embedding maps to lie in the set $\mathcal{C} = \{f : \mathbb{R}^D \to \mathbb{S}^{d-1}\}$, where we consider $d = 2, 3$, making the setup closer to the original NT-Xent loss in eq. (1). All experiments in the paper are conducted using one NVIDIA GeForce RTX 2080.

These experiments further illustrate how neural network optimization influences training dynamics. We also compare the behavior of vanilla gradient descent (assuming that $f$ is fixed at time $t = 0$ as in eq. (2)) with that of gradient descent through a neural network. Despite the change in output dimension, the same phenomenon persists: neural network training guides the dynamics toward stationary points that reflect the underlying clustering structure of the data. In contrast, vanilla gradient descent, which lacks the representational capacity of the neural network, remains insensitive to the data structure and fails to produce meaningful separation.

In Figure 6, we use a simple artificial dataset in $\mathbb{R}^2$ consisting of four clusters aligned along the $x$-axis, with each cluster generated from a Gaussian distribution centered at $(-1, 0)$, $(0, 0)$, $(1, 0)$, and $(2, 0)$, respectively. Points are colored according to their cluster labels, and arrows indicate the

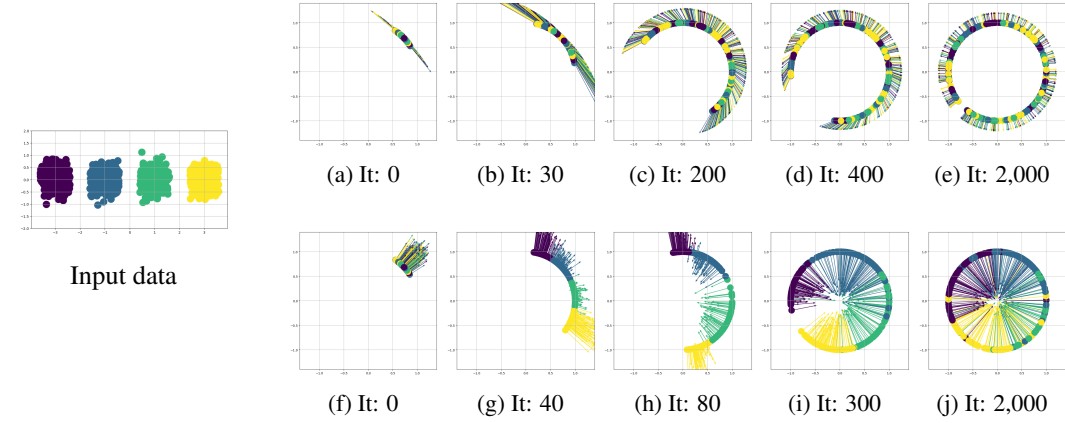

Input data

(a) It: 0    (b) It: 30    (c) It: 200    (d) It: 400    (e) It: 2,000

(f) It: 0    (g) It: 40    (h) It: 80    (i) It: 300    (j) It: 2,000

Figure 6: Comparison of the optimization process with and without neural network training using the simplified loss in eq. (12). In each figure, points are colored according to their cluster assignments based on the input data, and arrows denote the negative gradient. Row 1 shows optimization using vanilla gradient descent, where the distribution eventually becomes uniformly dispersed, disregarding the clustering structure of the input data. Row 2 shows optimization with neural network training, where the clustering structure becomes linearly separable in the early iterations. This outcome is consistent with Theorem 4.3.

direction of the negative gradient computed at each point. The key parameters are set as follows: $\tau = 0.2$, $\delta = 0.3$, and $n = 2{,}000$. We use stochastic gradient descent with a learning rate of $0.001$ to optimize the model. The training dynamics are examined using the simplified contrastive loss in eq. (12), comparing two optimization strategies: vanilla gradient descent and gradient descent through a neural network.

For vanilla gradient descent (Row 1 in Figure 6), the training dynamics show that the data points gradually spread until they form a uniformly dispersed distribution on a sphere, effectively erasing the initial clustering structure. This behavior aligns with Theorem 4.2, which asserts that the gradient of the loss function is independent of the input structure.

In contrast, neural network optimization produces linearly separable feature representations early in training, with clusters becoming distinctly separated by hyperplanes, thereby confirming Theorem 4.3. Despite random initialization, the contrastive loss gradient effectively guides the features so that the error terms of order $\delta$ remain negligible over short intervals.

Figure 7 compares the optimization processes with and without neural network training in both 2D and 3D, using different neural network architectures from that used in the main text and earlier figures. While the main text and preceding experiments use a simple one-hidden-layer neural network, Figure 7 employs a deeper 4-layer fully connected neural network. This demonstrates that the same phenomenon, emergence of clustering under neural network training, persists regardless of the network architecture.

The initial data distributions are shown in panels (a) and (l). The color of each point corresponds to its respective cluster. Rows 2 and 5 show the optimization with neural network training, starting from a random initial embedding and progressively revealing the clustering structure over time. In contrast, rows 3 and 6 display the optimization with vanilla gradient descent, where the feature distribution gradually converges to a uniform arrangement, disregarding the clustering structure present in the input data.

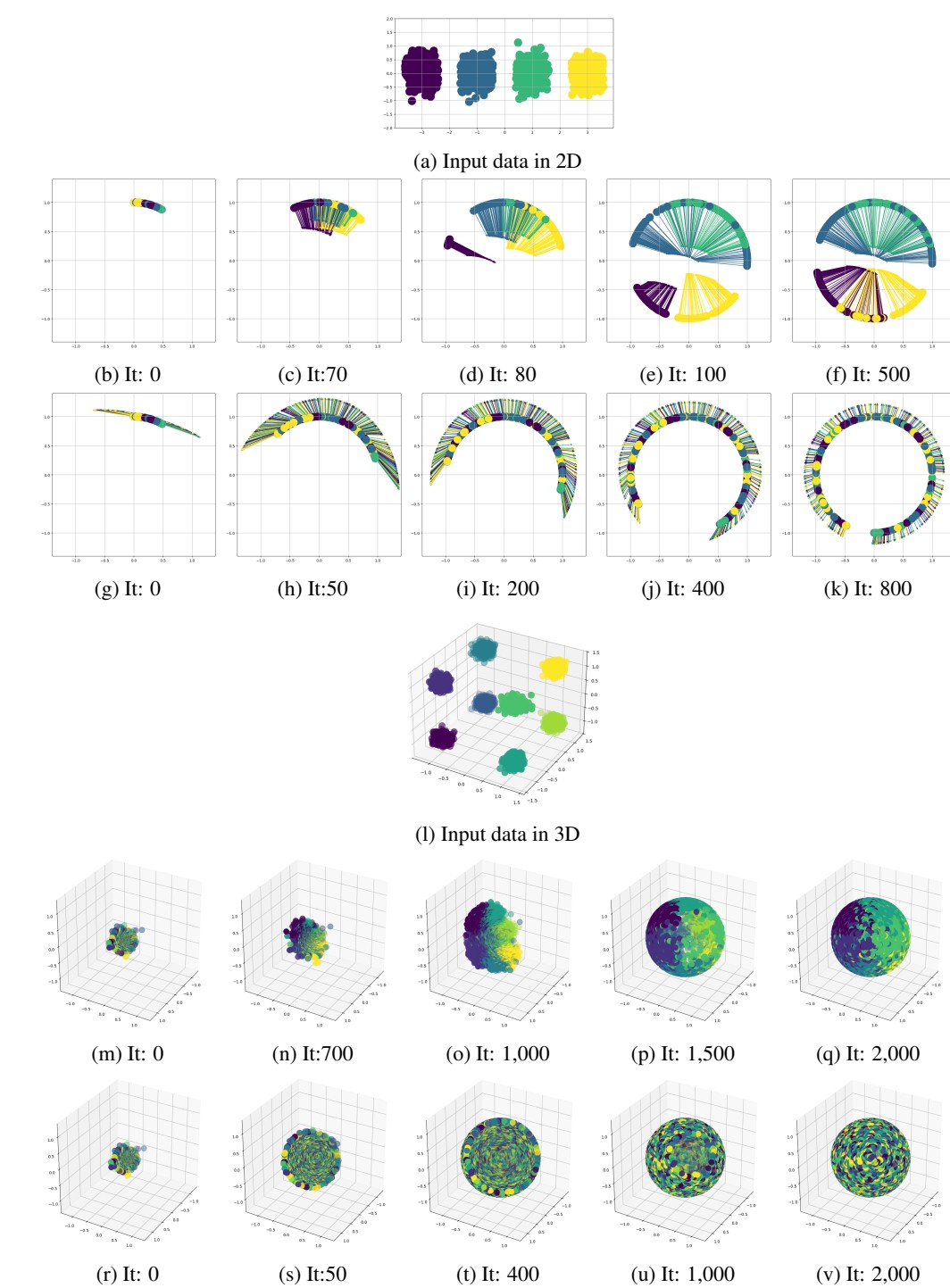

(a) Input data in 2D

(b) It: 0    (c) It:70    (d) It: 80    (e) It: 100    (f) It: 500

(g) It: 0    (h) It:50    (i) It: 200    (j) It: 400    (k) It: 800

(l) Input data in 3D

(m) It: 0    (n) It:700    (o) It: 1,000    (p) It: 1,500    (q) It: 2,000

(r) It: 0    (s) It:50    (t) It: 400    (u) It: 1,000    (v) It: 2,000

Figure 7: This experiment compares the optimization processes with and without neural network training in 2D and 3D, with the data distribution depicted in (a) and (l). A 4-layer fully connected neural network demonstrates consistent outcome as in Figure 6. Each point's color indicates its cluster. Rows 2 and 5 show optimization with neural network training, starting from a random embedding and gradually revealing the clustering structure. In contrast, Rows 3 and 6 illustrate the optimization process using vanilla gradient descent, which converges to a uniformly dispersed arrangement, disregarding the input data's clustering structure.