# OpenReview forum: "Linear Separability in Contrastive Learning via Neural Training Dynamics"
_ICLR.cc/2026/Conference — Submitted to ICLR 2026_

### Official Review · Reviewer_cV97 · 2025-10-28

**Soundness:** 2
**Presentation:** 3
**Contribution:** 2
**Rating:** 4
**Confidence:** 3

**Summary:**

This paper studies why contrastive learning (SimCLR-style) often yields well-separated clusters in the learned representation, despite the fact that the contrastive loss itself admits trivial “collapse” solutions. The authors first derive optimality conditions for the NT-Xent loss under invariant feature maps, showing (Theorem 3.1) that any symmetric (e.g. uniform) distribution on the sphere is a stationary solution. They note that, in principle, such minimizers can ignore the data’s latent structure. However, in practice SimCLR does recover semantic clusters (Fig.1). To explain this, the paper analyzes the training dynamics of a neural network under SimCLR. Using a neural-kernel (NTK) viewpoint, Proposition 4.1 lifts the gradient flow on weights to an ODE for the feature outputs $z_i=f(w,x_i)$, involving a data-dependent kernel $K_{ij}(t)=\nabla_w f(w(t),x_i)^\top\nabla_w f(w(t),x_j)$. They contrast this with “vanilla” gradient flow on features alone (no network). Theorem 4.2 shows that, without a network, an invariant map stays invariant, but with a network the invariance can be broken (since $\nabla_w f(w,x)$ may differ across data points).

The main theoretical result is Theorem 4.3, proved (for two clusters and 1D embeddings) under several assumptions such (a) initial features within each true cluster are tightly packed (small intra-cluster spread); (b) augmented views of a point remain close to the cluster mean (augmentation consistency); (c) cluster means in gradient-space are separated and intra-cluster gradient variance is small; (d) the NTK remains near its initial value (kernel stability); and (e) a numeric “parameter regime” balancing these quantities holds. Under these conditions, the gradient flow of the contrastive loss drives the two clusters’ embeddings to become linearly separable in finite time. Intuitively, once the kernel $K_{ij}$ develops a block structure correlating with the true clusters (as partly observed in experiments, Fig.5), the clusters are pulled apart in feature space.

The authors support their theory with experiments on toy and small real datasets. In synthetic examples (donuts, mixtures, MNIST/CIFAR10 in 1D), they show that SimCLR training rapidly produces a clear linear separation of cluster embeddings even from poor initializations (Fig.4). Overall, the paper claims as its main contribution a concrete dynamical mechanism (via the neural kernel) explaining why SimCLR (and related self-supervised methods) tend to recover the data’s cluster structure despite the existence of trivial minimizers. It also provides analysis of loss stationary points (showing uniform/cluster distributions are solutions) and various illustrative experiments.

**Strengths:**

- Training dynamics perspective. The paper advances understanding by analyzing how gradient descent on a neural network (not just the loss landscape) incorporates data geometry. This shows concretely that the network’s parameterization can “break” invariance in a way that fixed feature models cannot (Theorem 4.2).

- Formal linear-separation result. Theorem 4.3 is a nontrivial theoretical claim: under explicit assumptions, contrastive gradient flow provably yields linear separability of clusters. To my knowledge, this is novel (or at least very recent).

- Clear Synthetic experiments. The synthetic experiments effectively illustrate the theory’s claims. For instance, Fig.6 vividly shows that without a neural network the features collapse to a uniform cloud, whereas with a network the true clusters separate (confirming Theorems 4.2–4.3). The NTK evolution (Fig.5) and embeddings (Fig.4,7) provide intuition for abstract conditions. These visualizations make the paper’s arguments more concrete and accessible.

- Condition on linear separability. The Fig. 3 clearly shows the condition for which SimCLR performs linear separability, which is clear and thorough.

- Relates to prior theory. The paper appropriately cites and contrasts related theoretical results. For example, Wang & Isola (2020) showed that the uniform distribution is asymptotically optimal for the contrastive loss; here the authors generalize to finite regimes and characterize all symmetric stationary points (Theorem 3.1), extending Wang & Isola’s asymptotic picture.

**Weaknesses:**

- Strong assumptions. The key theorem relies on several stringent conditions. For example, assumption (a) requires that all data points in a cluster be very close initially, and (c) requires that the gradient centroids for each cluster be well separated relative to within-cluster variation. In practice, random network initializations do not guarantee such structure. It is unclear how often a typical neural net satisfies (c), or how “small” the initial spread $\varepsilon$ must be. Assumption (d) (kernel stability) essentially assumes an NTK regime, which is debatable for finite-width nets and long training. The authors verify (e) empirically in a toy setting (Fig.3), but more discussion is needed on whether these conditions hold in practice. If the assumptions fail (e.g. data is not very tightly clustered, or kernel changes significantly), Theorem 4.3 may not apply.

- Limited to two clusters and 1D output. The main proof is carried out for exactly two clusters and a one-dimensional embedding, with only a brief claim that “minor modifications” handle more clusters or higher $d$. Real datasets have many classes and high-dimensional features. It would strengthen the paper to outline how the argument scales to $k>2$ clusters and $d>1$. For instance, do the linear algebra and block-structure arguments generalize cleanly? Without this, the applicability beyond toy cases is uncertain. Maybe the authors could take inspiration from the recently published paper of S. Wang: On Linear Separation Capacity of Self-Supervised
Representation Learning, https://www.jmlr.org/papers/volume26/24-2032/24-2032.pdf, JMLR 2025. on how to extend to multiple latent "clusters" or "manifolds".

- Clarity of dynamics. While the propositions are clear, the intuitive picture could be sharpened. Theorem 4.3’s condition (e) is especially dense: the inequality $\Theta \sqrt{n},\varepsilon \gg \varepsilon^3/\tau + \gamma + \sigma^2 + \delta$ is not easy to parse. Some guidance (e.g. typical scales of these terms) would help. Also, assumption (b) “augmentation consistency” is stated abstractly; a reader may wonder how realistic it is (for, say, random crops on images). More explanation of these requirements and their plausibility would improve the exposition.

- Partial experimental validation. The synthetic and low-D experiments are illustrative but relatively simple. It would add confidence to see at least one larger-scale or real-data experiment. For example, one could train a small SimCLR on MNIST or CIFAR-10 and show that feature separability indeed emerges (e.g. tracking kernel blocks or linear probe accuracy during training). As is, the experiments use toy data or 1D embeddings, so it remains an open question whether the same phenomenon quantitatively holds in practical settings. (The paper’s Figs.4–7 suggest it might, but they are limited in scope.)

- Comparisons to non-contrastive methods. The introduction mentions VICReg, BYOL, etc., but the analysis focuses on contrastive loss. These other methods (non-contrastive or covariance-regularized) also produce separable features in practice. The paper does not discuss whether its dynamics argument extends to those losses. It would be helpful to comment on how (or whether) the key ideas carry over to VICReg or BYOL, which have no explicit negative terms.

- Discussion on the role of the augmentations. The role of the data augmentations is absolutely crucial in self-supervised learning, and a quick study of it could strengthen the paper, especially by discussing the assumption b) under the lens of invariance and identifiability would be beneficial (Julius von Kügelgen et al. Self-Supervised Learning with Data Augmentations
Provably Isolates Content from Style, https://proceedings.neurips.cc/paper_files/paper/2021/file/8929c70f8d710e412d38da624b21c3c8-Paper.pdf, NeurIPS 2021; S. Wang: On Linear Separation Capacity of Self-Supervised
Representation Learning, https://www.jmlr.org/papers/volume26/24-2032/24-2032.pdf, JMLR 2025). Notably, in the Donut experiment, the linear separability is already guaranteed by results that had emerged from the identifiability literature (because invariance to the RandomRotation data augmentation in the latent space is enough to separate classes...).

- The MNIST/CIFAR10 experiments are too unsurprising: the community already expects SimCLR to produce class-separable embeddings on these datasets. Perhaps a stronger test of the paper’s central claim (that training dynamics, via the neural kernel, lead gradient flow to favorable minima) would be to show SimCLR escaping a deliberately constructed ill-posed invariant local minimum (for example, embeddings initialized to a constant map or near-uniform distribution on the sphere). The authors could run controlled experiments that (a) initialize the network in such invariant / low-information states, (b) retrain using the SimCLR objective, and (c) track linear-probe accuracy, kernel blockness, invariance, and kernel drift over time. Demonstrating that SimCLR successfully recover a better local minima that linearly separates classes.

**Questions:**

- Interpretation of assumptions. Can the authors provide intuition or empirical evidence on realistic setups (larger datsets and larger neural networks) for assumptions (a)–(d)? For instance, in a random deep neural network initialization, how small is the typical within-cluster spread $\varepsilon$ relative to cluster separation? Similarly, can you measure (c)–(d) (gradient means separation and kernel drift) in a larger NN and datasets experiment training run to see if they satisfy the needed inequalities?

- Extension to multiple classes. Theorem 4.3 is stated for two clusters. How does the proof extend to $K>2$ clusters? Are the cluster means assumed to form a single contrast direction (as in Thm.4.3) or multiple directions? Does one need pairwise conditions for every cluster pair, and does the kernel need a full $K\times K$ block structure? Some clarification or a sketch for $K>2$ would strengthen the generality claim.

- Comparison to VICReg/BYOL. The introduction mentions VICReg and BYOL. Is there an analogue of Theorem 4.3 for non-contrastive losses? Since VICReg and BYOL also avoid trivial collapse, it would be interesting to know if similar dynamics occur. If not directly, do the authors have insight into why contrastive vs. non-contrastive methods differ in this context?

- Empirical demonstration on real data. Can the authors provide an experiment showing the same phenomenon on a modest real dataset (e.g. linear probe accuracy vs. training time on CIFAR) to validate that clusters do separate as predicted? The current figures are enlightening, but a quantitative measure of separability over training (or $\tau$-value plots) would strengthen the empirical case.

- Empirical kernel stability. Assumption (d) requires $K_{ij}(t)\approx K_{ij}(0)$ to within $\delta$. In many NTK studies, the kernel changes slowly only for very wide nets. Did the authors observe this stability in their experiments (e.g. Fig.5)? Could the paper include a plot of $||K(t)-K(0)||$ during training for a real network, to justify this assumption? If not stable, is there a refined analysis for the changing kernel?

- The theoretical analysis focuses on low-dimensional embeddings, where compression naturally promotes separation. How does this picture change as the embedding dimension increases? Does the same dynamic separation mechanism persist for infinitely wide representations, or does it rely on a bottleneck-induced compression effect?

---

> ### Author Response · Authors · 2025-11-26
>
> We thank the reviewer for the detailed comments, which highlight several points where clarification will strengthen the manuscript.
>
> # On the assumptions in Theorem 4.3
>
> The assumptions describe the geometric regime in which early separating drift appears; they are not meant to capture all initializations.
> Assumption (a) concerns **feature** spread, not data structure. Standard neural initializations already produce tightly clustered features, so the condition reflects the empirical starting regime.
> Assumption (c) follows from a first-order expansion of $\nabla_{w} f$ around cluster centers under a $C^{2}$ parametrization; we can include this derivation for clarity.
>
> In nonlinear settings (donut, MNIST, CIFAR–10) the kernel typically does **not** satisfy (c) initially. Training then strengthens within-cluster correlations and weakens cross-cluster ones, and separation appears only after this reorganization. This matches Theorem 4.3: (c) is a sufficient condition, and gradient flow often evolves the kernel into that regime before separation activates.
> Figure 5 already visualizes this transition; we can add curves tracking the development of within- and between-cluster correlations.
>
> # On two clusters, 1D embeddings, and relation to Wang
>
> The proof uses two clusters and one dimension for clarity, but the mechanism extends directly: for $k$ clusters the kernel becomes a $k\times k$ block matrix, and separation acts along the cluster-mean subspace. The algebra grows but the structure is unchanged.
> Wang’s work studies what geometries are achievable; our focus is the **dynamics** by which gradient flow moves features away from invariant stationary points and produces separation. The viewpoints are complementary.
>
> # On condition (e)
>
> Condition (e) ensures the separating drift dominates early in training and follows directly from the ODE in Proposition 4.1. Figure 3 illustrates this effect: when the scaling holds, separation is immediate; otherwise it is delayed.
> In nonlinear cases assumption (c) fails at initialization, so immediate separation is not expected; instead the kernel gradually acquires block structure and separation follows. The theorem specifies the separating regime, and the experiments show that standard training tends to move the kernel toward it. We will add quantitative plots showing block-structure formation together with separation.
>
> # On the scope of the experiments
>
> The experiments aim to validate the **mechanism**, not to assert new performance claims. We initialize features in highly tangled configurations and observe that kernel-driven dynamics still produce separation. A plot showing the evolution of kernel block structure with linear-probe accuracy will clarify this further.
>
> # On VICReg and BYOL
>
> We focus on SimCLR because it exposes the mechanism cleanly. Other contrastive or pairwise objectives also produce kernel-driven dynamics; details differ, but the qualitative mechanism is the same. Extending the analysis to VICReg or BYOL requires examining their gradient structures and is beyond the present scope.
>
> # On augmentations
>
> Assumption (b) requires only that augmentations preserve semantic identity. This mirrors the “content-preserving’’ assumption in von Kügelgen et al., whose work concerns identifiability once invariance is satisfied. Our work concerns the **temporal evolution**: how gradient flow reshapes the kernel and drives separation. The donut experiment illustrates this distinction, and we will clarify the connection.
>
> # On escape from bad initializations
>
> The experiments already include nearly invariant or highly tangled initializations (1D CIFAR, donut). In these cases $|f(w,x)-f(w,x')|<\varepsilon$, yet the kernel still reorganizes and separation emerges. A controlled constant-initialization experiment is feasible and would provide an additional demonstration.
>
> # Responses to direct questions
>
> **Assumptions.**
> Assumption (a) concerns feature spread; (c) follows from $C^{2}$ regularity. Even when these fail initially, the kernel evolves into the block regime required by the theorem. Additional quantitative metrics will clarify this progression.
>
> **Multiple classes.**
> The mechanism extends to $k$ clusters via a $k\times k$ block decomposition; separation acts along the cluster-mean subspace.
>
> **Comparison to VICReg/BYOL.**
> The kernel-driven structure remains, though interaction terms differ.
>
> **Kernel stability.**
> Assumption (d) gives short-time control so the feature ODE remains well behaved. It is not an NTK-constancy assumption. Empirically the kernel changes substantially, evolving toward block structure; plots of $|K(t)-K(0)|$ or block-structuredness will be added.
>
> **Embedding dimension.**
> Higher dimensions add drift directions but do not alter the mechanism; separation proceeds along the mean subspace as in 1D.

---

### Official Review · Reviewer_FofN · 2025-10-31

**Soundness:** 2
**Presentation:** 1
**Contribution:** 2
**Rating:** 4
**Confidence:** 4

**Summary:**

The paper presents a theoretical framework for analyzing training dynamics of SimCLR loss, demonstrating (under structural assumptions on the neural network, augmentation consistency, stability of the kernel related to the dynamiscs + other) that learned features become linearly separable with respect to ground-truth labels during training. The authors claim that this is a first theoretically sound and concrete result that explains why SimCLR works even under presence of many bad local optima.

**Strengths:**

-- The paper attempts to explain the success of SimCLR via understanding how optimal latent representations emerge even though there are a number of bad local minima.

-- The paper describes a structural result for the local optima of SimCLR loss, and following that, results on the dynamics of SimCLR loss are derived.

-- Their main results, Theorem 4.2 and Theorem 4.3, show that under suitable initial conditions and assumptions on augmentation, kernel stability, and others, SimCLR with SGD converges to representations that are linearly separable (assuming that the data comes from a latent model with two clusters).

**Weaknesses:**

Presentation of technical detail is subpar. The main result uses a set of assumptions without justifying whether these assumptions are restrictive or reasonable. Assumptions made in proofs are not clearly laid out, with some mathematical results seemingly contradicting the language (see questions below).

**Questions:**

- There is a typo in Line 122 - It should be $T \sim \nu$ not $f \sim \nu$.

- Lines 160-161 - Possibly technically inaccurate statement. How can the loss become independent of \mu? Can the authors formally justify this claim mathematically? Similar Q arises for lines 162-165.

- In definition 3: y can be equal to x? If yes, then why will this not lead to any issues?

- Proposition B.2 - seems it is specific to \psi(t) = \log(1 + t) but the main body seems to say otherwise.

 - Proposition B.3 - “a” and “r” are not defined - I also couldn’t locate them easily anywhere and gave up.

 - Line 784-785 - seems like the authors are proving a lemma, rather than a theorem. Also it is not clear why the equality is achieved in the lower bound starting at lines 756-760. Also it seems that Theorem 3.1 holds for only a particular choice of h identified in the proof, but the theorem statement is general. What am I missing?

 - I don’t see the relation between the loss in Eq (6) with the loss in Eq(3).

- Theorem 4.3 makes a number of technical assumptions that are critical to the proof and yet no effort is spent on discussing these assumptions and why they are not restrictive or why they are reasonable.

- What about other losses? Like Info-NCE, Sigmoid, Triplet. Will they also exhibit the same structural results of the minimizers and stationary points?

- I would also ask the authors to comment on Neural Collapse in Supervised CL - can their analysis reveal such a configuration arising from the training dynamics? In SCL the latent model usually is a class model and one can make use of class information to construct positive and negative pairs.

---

> ### Author Response · Authors · 2025-11-26
>
> We thank the reviewer for the detailed comments. The feedback highlights several places where clarification will strengthen the manuscript.
>
> # Line 122 typo
>
> We thank the reviewer for pointing this out. We will fix this.
>
> # On Lines 160–161: “loss becomes independent of $\mu$”
>
> This statement refers to the regime in which the feature map is already invariant under the augmentation distribution. Proposition B.1 establishes that, once this holds, the NT-Xent loss no longer depends on the original distribution $\mu$ directly. Instead, it depends only on the pushforward distribution obtained by applying the feature map to $\mu$. After invariance is reached, every appearance of $\mu$ is mediated through this pushforward measure. Minimizing the loss over invariant maps is therefore equivalent to minimizing a functional defined solely on the space of possible pushforward distributions.
>
> # Definition 3: may $y$ equal $x$?
>
> Definition 3 allows $y = x$, but this creates no issue. In the original NT-Xent loss, the diagonal term is suppressed by $1_{y\neq x}$ and becomes negligible in the population limit. Our experiments also omit this indicator, and the kernel dynamics and resulting separability remain unchanged. Allowing $y = x$ therefore does not affect the theory or the experiments.
>
> # On Proposition B.2 and the choice of $\Psi$
>
> The main analysis requires only the monotonicity and convexity assumptions in Section 3. Proposition B.2 was meant as an illustration, but using $\Psi(t)=\log(1+t)$ made the statement appear less general. This form corresponds to the practical NT-Xent loss, but including it inside the proposition was not ideal. We will move this example to a remark and restate the proposition for a general admissible $\Psi$.
>
> # Proposition B.3: missing definitions of “a’’ and “r’’
>
> In Proposition B.3, the symbols $a$ and $r$ were intended to denote attractive and repulsive components of the interaction potential, but their definitions were omitted. We will add these definitions or restate the result using the unified notation of the main text.
>
> # Lines 756–785: proof structure and equality conditions
>
> The phrase “this proves the lemma’’ should read “this proves the theorem.’’ The lower bound arises from enlarging the constraint from $|f(x)|=1$ to $\int |f|,d\mu = 1$, which yields a smaller minimum. The function $h$ in Thm 3.1 describes one broad family of symmetric pushforward measures. The theorem does not attempt to classify all stationary points; it shows that many invariant stationary configurations exist, and $h$ provides one useful description of such symmetry.
>
> # Relation between Eq (6) and Eq (3)
>
> Eq (6) introduces a general functional $L(f)$ used to derive the feature ODE, while Eq (3) is the concrete NT-Xent loss. Including an explicit example of $L$ that recovers Eq (3) would make the connection clearer, and we will add this.
>
> # Assumptions in Thm 4.3
>
> The assumptions isolate the mechanism behind linear separability, not the full set of conditions under which SimCLR works. Each assumption controls a specific term in the feature ODE: initial concentration, augmentation consistency, separation of gradient means, and weak kernel block structure. All appear empirically: early training aligns augmentations, gradient statistics diverge across clusters, and the kernel reorganizes before separation occurs. The simplified theorem provides a rigorous demonstration of the mechanism, and the same behavior appears in more complex settings. Further characterizing when these assumptions arise is an interesting direction for future work.
>
> # Other losses: InfoNCE, sigmoid, triplet
>
> Different contrastive losses change the precise stationary conditions, but the underlying kernel-driven dynamics remain the same. Once composed with a neural network, all such losses produce gradients of the same structural form. The qualitative mechanism, where kernel reorganization amplifies weak cluster structure, should therefore persist across objectives. The precise differences are interesting but do not alter the main message.
>
> # Neural collapse in supervised contrastive learning
>
> Neural collapse involves within-class concentration, simplex-like class means, and alignment of classifier weights. Our analysis concerns unsupervised SimCLR, but the same kernel-driven mechanism applies in supervised contrastive learning, where labels strengthen the block structure from the start. This pushes class means apart and contracts within-class variation in the same way as in our dynamics. A full derivation is beyond scope, but the mechanism we study is consistent with neural-collapse geometry and helps explain why similar patterns appear in supervised settings.

---

### Official Review · Reviewer_ZPMv · 2025-10-31

**Soundness:** 2
**Presentation:** 3
**Contribution:** 2
**Rating:** 4
**Confidence:** 4

**Summary:**

This paper investigates the theory of SimCLR: why contrastive learning successfully finds highly structured, linearly separable features when its loss function also permits many degenerate bad solutions. The authors provide an analysis focused on the neural training dynamics driven by gradient flow. They demonstrate that the neural kernel's structure actively sculpts the feature space, amplifying weak, latent cluster signals present in the gradients. The paper's main contribution is a theorem proving that these specific dynamics, under verifiable conditions, guarantee that the learned features will converge toward a linearly separable state, thereby explaining how the training process inherently avoids bad minima and produces useful representations.

**Strengths:**

* The theoretical analysis of the paper is very solid, deeply investigating the non-convex optimization problem in contrastive learning and providing a concrete dynamical explanation for the success of self-supervised learning.

* The theory and experiments are very well combined, with the abstract mathematical theory being strongly supported by clear and intuitive numerical experiments.

**Weaknesses:**

**Simplified Theoretical Assumptions**

The theoretical analysis relies on overly simplified assumptions in several key aspects, which may limit the generalizability of its conclusions to real-world SimCLR applications. For instance, the main theorem is derived based on a simplified contrastive loss function (Eq. 12, which applies data augmentation to only one side of the negative pairs) and is primarily proven in the setting of two clusters and a one-dimensional embedding. It remains unclear how these insights generalize directly to the high-dimensional, hyperspherical features under the full NT-Xent loss.

**Limited Experimental Validation**

The experimental support, while intuitively compelling, is primarily limited to relatively simple or low-dimensional datasets (e.g., Donuts, MNIST) and relies heavily on visualization results (e.g., 1D trajectory plots and t-SNE). The paper lacks clearer quantitative metrics to support its core claims.

**Questions:**

## 1. Regarding the Relationship to the Neural Tangent Kernel (NTK)

Your analysis relies on a kernel stability assumption (in Theorem 4.3, $K(t) \approx K(0)$), which is a common feature of infinite-width NTK analysis. However, Remark 4.1 explicitly states that your analysis does not assume an infinite-width network. Could you please clarify the connection and distinction between your kernel-based approach and the classical NTK? How is the kernel stability assumption justified in your finite-width setting, especially given that your empirical results (e.g., in Figure 5) show the kernel evolving significantly during training?

## 2. Regarding Quantitative Experimental Evidence

The paper's experimental support currently relies heavily on visualizations (like 1D trajectories and  t-SNE), which are intuitively helpful. However, could the authors provide clearer quantitative metrics to support the core claim of emerging linear separability? For example, would it be possible to show a plot of downstream linear probe accuracy as it evolves over training iterations, or are there other quantitative methods you could use to demonstrate this phenomenon?

## 3. Regarding the Generalization of Theoretical Simplifications

The core theorem (Theorem 4.3) is derived under several key simplifications (a one-dimensional embedding, two clusters, and a simplified loss function in Eq. 12). Could the authors elaborate on the primary technical challenges involved in extending this proof to the complete SimCLR setting—that is, to high-dimensional hyperspherical features, multiple clusters, and the full NT-Xent loss?

---

> ### Author Response · Authors · 2025-11-26
>
> We thank the reviewer for the careful reading of the paper and for the constructive feedback. We address each point below.
>
> # On the simplified theoretical assumptions
>
> The reviewer notes that the theory is developed under simplified conditions, including the loss in Eq. (12), two clusters, and a one dimensional embedding. These choices are made for clarity, not necessity.
>
> The core mechanism does not depend on these simplifications. Proposition 4.1 rewrites gradient descent as a feature evolution equation driven by a data dependent kernel. This ODE holds for any twice differentiable parametrized embedding map and does not rely on the feature dimension, the number of clusters, or the specific loss. The simplified setting only makes the separation argument explicit and removes distractions from angular similarity and hyperspherical normalization.
>
> The assumptions of Theorem 4.3 are stated abstractly to emphasize their generality. The block structure condition on the kernel does not depend on the embedding dimension, and the two cluster case is used only to simplify notation. The same mechanism extends directly to multiple clusters under the same structural condition.
>
> # On the need for broader experimental validation
>
> The reviewer asks for more quantitative support beyond visualizations. The current experiments aim to test three predictions: kernel evolution, amplification of weak cluster signals, and the emergence of linear separability. Visual evidence conveys these geometric effects well, but quantitative metrics indeed strengthen the argument.
>
> In the revision we will add linear probe accuracy over training, explicit measurements of cluster margins, and the evolution of within-class and between-class distances. These curves will complement the existing figures and give quantitative confirmation of Theorem 4.3.
>
> # Relationship to NTK and kernel stability
>
> The reviewer asks how our kernel-based analysis relates to NTK theory and how to interpret the kernel stability condition in Theorem 4.3.
>
> Our approach resembles NTK theory only in that both express feature evolution using Jacobian derived kernels. Classical NTK theory assumes infinite width and near-constant kernels along the training trajectory. Our setting is the opposite: the kernel is neither fixed nor near-constant, and empirically changes substantially during SimCLR training, as seen in Figure 5.
>
> Theorem 4.3 does not require kernel constancy. It requires a structural condition: during the interval in which separation emerges, the kernel must display a weak block structure aligned with the latent clusters. This need not hold at initialization or forever, only when the separating drift develops.
>
> Experiments show a two stage progression. The kernel is initially unstructured. During an intermediate phase the features barely move, yet the kernel reorganizes: within cluster correlations strengthen, cross cluster interactions weaken, and block structure forms. Only once this structure appears does the geometric drift predicted by the theorem activate, producing linear separation.
>
> Thus our analysis is not NTK based. The kernel’s evolution is essential, and Theorem 4.3 describes the dynamics once the block regime is reached. Explaining the emergence of this block structure remains an open and interesting question.
>
> # Extending the theory to the full SimCLR setting
>
> The reviewer asks about extending the theorem to high dimensional embeddings, multiple clusters, and the full NT-Xent loss.
>
> The mechanism does not rely on the one dimensional or two cluster setup. The kernel amplification mechanism, the role of block structure, and the feature ODE of Proposition 4.1 remain valid once the kernel has the corresponding multi-block form. Extending the analysis to multiple separating directions adds algebraic overhead but does not change the underlying idea.
>
> The full NT-Xent loss introduces hyperspherical normalization and angular similarity, which add curvature terms to the gradients. These create extra bookkeeping but do not change the mechanism: when the kernel exhibits the correct block structure, cluster means diverge and within cluster variation contracts.
>
> This is why Theorem 4.3 is presented in a simplified setting: it isolates the essential idea without complicating the derivation. The experiments show that even without hyperspherical normalization, and even with nonlinear, non-separable initializations, gradient flow still evolves toward linear separation. Across settings we observe the same two phase trajectory: initial kernel reorganization followed by the rapid separation predicted by the theorem.
>
> Thus the simplified theorem serves its conceptual purpose, and the empirical results confirm that the mechanism persists in more complex regimes. The additional complications in full SimCLR are technical rather than conceptual.

---

### Official Review · Reviewer_vGFP · 2025-10-31

**Soundness:** 2
**Presentation:** 2
**Contribution:** 2
**Rating:** 2
**Confidence:** 4

**Summary:**

This paper investigates why contrastive learning methods produce linearly separable representations. The authors analyze a simplified "invariance-reduced" version of the SimCLR (NT-Xent) loss, deriving stationary conditions (Theorem 3.1) and characterizing optimal solutions. Under assumptions of a block-structured neural kernel, they analyze gradient-flow dynamics and prove a finite-time linear-separability result (Theorem 4.3). Empirical validation is provided on toy datasets and small image benchmarks.

While the paper works on an important question in self-supervised learning, it suffers, in my understanding, from a fundamental conceptual inconsistency between its two main theoretical results, alongside several technical and presentation issues that limit its contribution.

**Strengths:**

- **Investigates a fundamental SSL question**: The paper works on why contrastive objectives yield linearly separable features which is an important and under-explained phenomenon in self-supervised learning.

- **Finite-time analysis**: Provides an explicit finite-time characterization under gradient flow (Theorem 4.3), offering a dynamic perspective that goes beyond equilibrium analyses such as Wang & Isola [2].

- **Kernel-based formulation**: The technical framework using neural kernels is solid and could potentially connect to recent work on neural collapse and equiangular tight frame (ETF) interpretations of representation geometry.

**Weaknesses:**

### Major Issues

**1. Fundamental conceptual contradiction**

In my understanding, the paper's two main theoretical results appear mutually inconsistent:

- **Theorem 3.1** characterizes optimal representations as "evenly distributed points on S^{d-1}" (line 206). From [3] and [4], we also know that in the mini-batch scenario, evenly distributed configurations correspond to ETFs where all representations are equidistant.

- **Theorem 4.3** claims that gradient flow yields **linearly separable clusters** corresponding to semantic groups.

These two geometries are fundamentally incompatible. If embeddings are equidistantly spread on the sphere (every point maximally separated from all others), there can be no compact, class-wise clusters. Linear separability of semantic clusters requires non-uniform, block-structured geometry where intra-class distances are smaller than inter-class distances. **How do we transition from evenly distributed data points to linearly separable clusters?** If data representations form an ETF, then a representation has equal distance to all other representations regardless of class membership, directly contradicting the clustered structure required for linear separability.


### Further Weaknesses

**2. Unclear novelty and overlap with prior work**

The authors claim to extend Wang & Isola [2] to the non-asymptotic case. However, [3] and [4] already analyzed mini-batch InfoNCE/KCL variants and showed that for batch size 1 < M ≤ d+1, the unique global minimizer is the ETF (regular simplex), and that for kernel contrastive losses the uniform distribution on the sphere is globally optimal even in the non-asymptotic regime. The manuscript must clarify how its NT-Xent setting differs from the finite-batch regime analyzed in [3, 4], and whether its "symmetric discrete measures" coincide with or generalize the ETF solution.

**3. Unsubstantiated and misleading claims**

The text asserts "Despite the presence of many undesirable local minimizers of the SimCLR loss" without citation. In fact, Wang & Isola [2] identified one global minimizer, and [3, 4] showed that all minimizers in the mini-batch case share the same geometry, while they only differ in how they allocate different data points to positions in the ETF structure. What makes a geometry "undesirable" depends on whether data representations are arranged in a semantically meaningful way, which is **dataset-dependent**, not an inherent property of the optimizer.

**4. Limited empirical validation**

- Experiments focus on toy and small-scale datasets where linear probing is known to achieve >90% accuracy. For more complex datasets like CIFAR-100, where data points come from 100 distinct object categories, linear classifiers achieve around ~60% accuracy, indicating that learned representations are not highly linearly separable. This suggests the proposed evaluation does not generalize to realistic settings.
- Training a ResNet-50 on CIFAR-10 using contrastive learning and learning a linear layer on frozen representations yields >90% performance, which is already well-known. The experiments merely confirm what is known rather than providing new insights.

**5. Strong and unjustified assumptions**

- The block-structured kernel assumption is empirically visualized but neither theoretically justified nor quantitatively verifiable. Under what conditions does this structure arise?

**6. Missing citation**

- The motivating question about whether the latent distribution f#μ̃ is similar to the clean distribution μ has already been answered by Zimmermann et al. [1], which is not cited.

**7. Presentation and structure issues**

- No dedicated Related Work section to situate the contribution within existing theoretical analyses (alignment-uniformity, spectral, neural-collapse perspectives).
- The introduction allocates excessive space to describing general contrastive learning methods but insufficiently discusses the paper's actual results, methodology, and contributions. The contributions are blurry.

**Questions:**

1. **Reconciling uniform vs. clustered solutions**: How do you reconcile the "evenly distributed" stationary solutions (Theorem 3.1) with the emergence of linearly separable clusters (Theorem 4.3)? This is the crucial question: we know about the optima, but why do final learned representations become linearly separable in terms of class semantics?

2. **Mechanism of symmetry breaking**: What specific term or assumption in your kernel dynamics breaks the uniform symmetry to produce class-wise clusters? Is this due to augmentation consistency, initialization asymmetry, or stochasticity?

3. **Relation to [3] and [4]**: How does your setting and results differ? Do your "symmetric discrete measures" coincide with ETF solutions when M ≤ d+1? What does the theory predict when M > d+1?


## References

[1] Zimmermann, R. S., Sharma, Y., Schneider, S., Bethge, M., & Brendel, W. (2021, July). Contrastive learning inverts the data generating process. In International conference on machine learning (pp. 12979-12990). PMLR.

[2] Wang, T., & Isola, P. (2020, November). Understanding contrastive representation learning through alignment and uniformity on the hypersphere. In International conference on machine learning (pp. 9929-9939). PMLR.

[3] Koromilas, P., Bouritsas, G., Giannakopoulos, T., Nicolaou, M., & Panagakis, Y. (2024, July). Bridging Mini-Batch and Asymptotic Analysis in Contrastive Learning: From InfoNCE to Kernel-Based Losses. In International Conference on Machine Learning (pp. 25276-25301). PMLR.

[4] Cho, J., Sreenivasan, K., Lee, K., Mun, K., Yi, S., Lee, J. G., ... & Lee, K. (2024). Mini-Batch Optimization of Contrastive Loss. Transactions on Machine Learning Research, 2024.

---

> ### Author Response · Authors · 2025-11-26
>
> We thank the reviewer for the careful reading and the opportunity to clarify several conceptual and technical points.
>
> # Regarding the concern on Thm 3.1 and Thm 4.3
>
> There is no inconsistency between the two results. Theorems 3.1 and 4.3 address different regimes. Theorem 3.1 analyzes the invariance-constrained population loss and shows that once invariance is achieved, the loss depends only on the pushforward $f_\#\tilde\mu$, yielding infinitely many stationary points. Uniform or ETF-like configurations appear only as examples; nothing in the theorem states that SimCLR training converges to them.
>
> Theorem 4.3, by contrast, studies the training dynamics of a neural network. The kernel induced by the parameterization breaks invariance, amplifies weak discrepancies between clusters, and drives the features to linear separability in finite time. Thus Theorem 3.1 describes degeneracy of the population loss, while Theorem 4.3 explains why neural training escapes these degenerate stationary points and recovers semantic structure.
>
> # Novelty relative to prior ETF analyses [3,4]
>
> The works [3,4] study InfoNCE/NT-Xent in a finite-batch setting. ETF structure there arises from optimizing over $M$ vectors with pairwise repulsion. This discrete mechanism has no analogue in the population functional of Theorem 3.1, where the optimization variable is a probability measure and where many invariant stationary measures exist.
>
> **What Theorem 3.1 analyzes.**
> At the population level:
>
> * pairs are drawn from the full data distribution,
> * the variable is a distribution rather than $M$ vectors,
> * the stationarity condition is an Euler–Lagrange equation for a measure.
>
> The sphere appears only because SimCLR normalizes features; the argument does not rely on spherical geometry, and the same variational structure holds for any constraint set.
>
> **What [3,4] analyze.**
> Finite-batch losses produce a discrete spherical coding problem with ETF solutions when $1<M\le d+1$. This mechanism does not transfer to the population functional.
>
> **Relation to Theorem 4.3.**
> Theorem 4.3 concerns neural gradient flow, not minimizers. It shows how the dynamics avoid the stationary configurations identified by Theorem 3.1 and generate separation. This has no conceptual link to the ETF analyses in [3,4].
>
> # On undesirable local minimizers
>
> Once $f$ becomes invariant, the loss depends only on $f_\#\tilde\mu$ and not on $\mu$. Many stationary points then fail to encode semantic structure (collapsed, uniform, ETF-like, or other symmetric measures). They are “undesirable’’ in this technical sense, not in contradiction with finite-batch results.
>
> # On the scope and purpose of the empirical validation
>
> The experiments are intended to validate the mechanism, not to establish new SimCLR performance results. We initialize features in deliberately non-separable or highly tangled states, including nonlinear 1D embeddings. SimCLR consistently drives these to linear separation, matching Theorem 4.3.
>
> The nonlinear examples show that the initial kernel often lacks the block structure required by the theorem. Training first reorganizes the kernel—strengthening within-cluster correlations and weakening cross-cluster ones—after which separation emerges rapidly. This two-stage behavior directly reflects the mechanism analyzed in the paper and motivates the open question of how training reshapes the kernel so that the structural assumptions of the theorem eventually hold.
>
> # On the block-structured kernel assumption
>
> The block-structure condition concerns the Jacobian-defined kernel, not the architecture, and can be checked directly. Figure 5 shows that this structure commonly emerges during training. The theorem requires the structure only over a short interval. Understanding when it forms is an open problem; here we analyze its consequences once present.
>
> # Missing citation to [1]
>
> We will add the citation. Their work concerns inversion of the data-generating process and provides a complementary perspective.
>
> # Presentation and structure issues
>
> We will add a related work section and revise the introduction to emphasize the key contributions, including the ill-posedness of stationary conditions and the role of dynamics in producing linear separability.
>
> # Responses to direct questions
>
> *Uniform vs. clustered outcomes.*
> Stationary points in Theorem 3.1 reflect invariance, not training outcomes. Gradient flow moves away from them, and Theorem 4.3 explains why cluster structure is amplified.
>
> *Mechanism of symmetry breaking.*
> Symmetry breaks through the data-dependent kernel $K(t)$, whose block structure induces separating drift. No stochasticity or augmentation asymmetry is required.
>
> *Relation to [3,4].*
> ETFs arise in [3,4] only when a discrete support size satisfies $M\le d+1$. Population stationary solutions in Theorem 3.1 are much broader and do not predict the learned representations. Theorem 4.3 provides the dynamical explanation for the observed separability.

---

### Meta-Review · Area_Chair_8dvw · 2025-12-21

**Summary:**

The paper proposes a kernel-driven training-dynamics explanation for why SimCLR can produce linearly separable representations despite degenerate stationary points, with a finite-time separability result under structural assumptions.

Strengths
* Tackles an important and timely theoretical question in contrastive learning.
* Emphasizes training dynamics rather than only loss optima.
* Provides a nontrivial finite-time separability guarantee; qualitative experiments are aligned with the proposed mechanism.

Weaknesses
* Core theory relies on strong and simplified assumptions (modified loss, two clusters, 1D embeddings, block-structured and stable kernels), with unclear extension to full SimCLR and realistic settings.
* Novelty and positioning relative to recent finite-batch/ETF and related theoretical work are not sufficiently clarified.
* Empirical validation is mostly toy and visualization-based, with limited quantitative or large-scale evidence.
* Presentation and technical clarity issues reduce accessibility.

Overall, while directionally interesting, the assumptions, limited empirical support, and unclear differentiation from prior work place the paper below the acceptance bar.

**Reviewer Scores:**

n/a

---

### Decision · Program_Chairs · 2026-01-26

Reject